
# Glacier detachments and rock-ice avalanches in the Petra Pervogo range, Tajikistan (1973–2019)

Silvan Leinss[1], Enrico Bernardini[1], Mylène Jacquemart[2], and Mikhail Dokukin[3]

[1]Institute of Environmental Engineering, ETH Zürich, Switzerland.
[2]Cooperative Institute for Research in Environmental Sciences, University of Colorado, Boulder, United States
[3]High-Mountain Geophysical Institute, Nalchik, 360030, Russia

**Correspondence:** leinss@ifu.baug.ethz.ch, enricobe@student.ethz.ch

**Abstract.** Glacier detachments are a rare phenomenon of glacier instability, whereof only a handful have been documented to date. Common to all known cases are the large detached volumes of many million cubic meters of ice and long runout distances. Recently, two detachments of smaller size were observed in the Petra Pervogo range, north west of the Pamir mountains, Tajikistan. Using a variety of satellite images, we identified in total 9 detachments and several ice and rock avalanches which
occurred in four different catchments between 1973 and 2019. The avalanche run out distances vary between 2 and $19\,\mathrm{km}$ and detached volumes range between 2 and $8.8 \times 10^6\,\mathrm{m}^3$. Seven out of nine detachments occurred between July and September in years with temperature above the past 46-years trend. No active glacier surge was observed immediately before detachment, but elevation model (DEM) differences indicate a surge-like behavior about 10 years before the two largest detachments. Instead, one glacier retreated before detachment while the other remained stagnant before increased sliding pronounced the impending
detachment. To put results into a regional context, we analyzed DEM differences over the entire Pamir range and found 237 surging glaciers, predominantly in the north-western part where soft and fine-grained rock-types are common. We are confident that no major events were missed due to lack of satellite data, because destroyed vegetation remains visible in the normalized difference vegetation index (NDVI), several years after large mass flows, e.g. about 10 years for the Kolka-Karmadon rock-ice avalanche. From the large number of detachments which occurred under very similar conditions we conclude that rising
temperatures combined with soft, fine-grained sediments are very critical components favouring the detachment of entire valley glaciers.

## 1 Introduction

### 1.1 Glacier detachments

Glacier detachments, where a large volume of a valley glacier decouples from its bed and results in subsequent high-velocity
travel of the detached ice mass (Evans and Delaney, 2015), are extremely rare and only a handful of them have been identified to date. Glacier detachments occur at a relatively low slopes around between 10° and 20° and precursors show similarities to glacier surges where a glacier's velocity increases by one or two orders of magnitude (Quincey et al., 2011), however, without detachment. Glacier surges are favoured by a climatic envelope and occur predominantly for rather long glaciers of





low slope (Sevestre and Benn, 2015). In contrast to detachments, ice or glacier avalanches, originating from steep headwalls
or hanging glaciers (Evans and Delaney, 2015), are much more frequent. For both mass flows, potential energy is transformed
into kinematic energy and into frictional heat. Frictional heat, and in additionally entrained sediments (Moore, 2014, Sect.
5.2.2), increase the liquid water content and makes the resulting mass flows, sometimes transformed to debris or mud flows,
highly mobile (Schneider et al., 2011; Evans and Delaney, 2015; Davies, 1982). The high mobility leads to much longer run
out distances than pure snow or rock avalanches (Schneider et al., 2011), potentially reaching inhabited areas (Petrakov et al.,
30  2008).

The reasons for detachments are not completely understood, but some factors seem to play a major role: some detached
glaciers presented a surge-like behaviour before detachment (Kääb et al., 2018) and one of the responsible mechanisms behind
surging appears to be increased water pressure which enhances basal sliding (Kamb et al., 1985; Harrison and Post, 2003;
Clarke et al., 2011). It is suspected that climate change increases the amount of meltwater and may thus favour development of
instabilities (Jacquemart et al., 2020). Another potential factor leading to instabilities are presumed to be soft bedrock litholo-
gies for which ice-sediment mixtures which can show "profound weakening at temperatures closer to melting" (Moore, 2014).
Soft sediments have been found for all of the probably best-studied events, the Kolka-Karmadon rock-ice avalanche (Droby-
shev, 2006; Huggel et al., 2005; Evans et al., 2009), the Aru Co twin glacier collapse (Kääb et al., 2018; Gilbert et al., 2018), and
the Flat Creek detachments (Jacquemart et al., 2020; Jacquemart and Loso, 2019). Despite different terminology, all of these
events would be classified as glacier detachments according to Evans and Delaney (2015). Together with comparable events
reported from China and Argentina (Paul, 2019; Falaschi et al., 2019), these glacier detachments raise the question of whether
such events might be more common than previously thought, and whether they might happen more often considering rising
global temperatures. While similarities between past events have provided a baseline understanding of the conditions promot-
ing, and factors triggering glacier detachments, their interactions and significance are still largely unknown. Yet precisely this
understanding needs to be improved to provide a robust assessment of these hazards.

The aim of this work is to provide an inventory of a series of glacier detachments and ice avalanches which occurred in
the Petra Pervogo Range, Tajikistan, between 1973 and 2019. We use this inventory to put the events in context with local
geology, climatic conditions, regional distribution of surge-type glaciers, and individual glacier's surging history. To build the
inventory we analyze a multiplicity of satellite imagery including the entire Landsat archive and investigated the usefulness of
normalized difference vegetation index (NDVI) to detect glacier detachments retroactively, if they occurred during years with
poor satellite coverage.

## 2   Study site

The Petra Pervogo range (also called Peter the First or Peter the Great range) is situated in central Tajikistan, north-west
of the Pamir mountain system. It extends to the east with Moscow Peak (6785 m) as the highest peak and is bordered by the
Surkhob and Obikhingou river to the north and south, both draining into the Vaksh river at the western end of the Petra Pervogo
range. In the Western Petra Pervogo range, shown in Fig. 1, we identified four catchments which showed repeated large mass




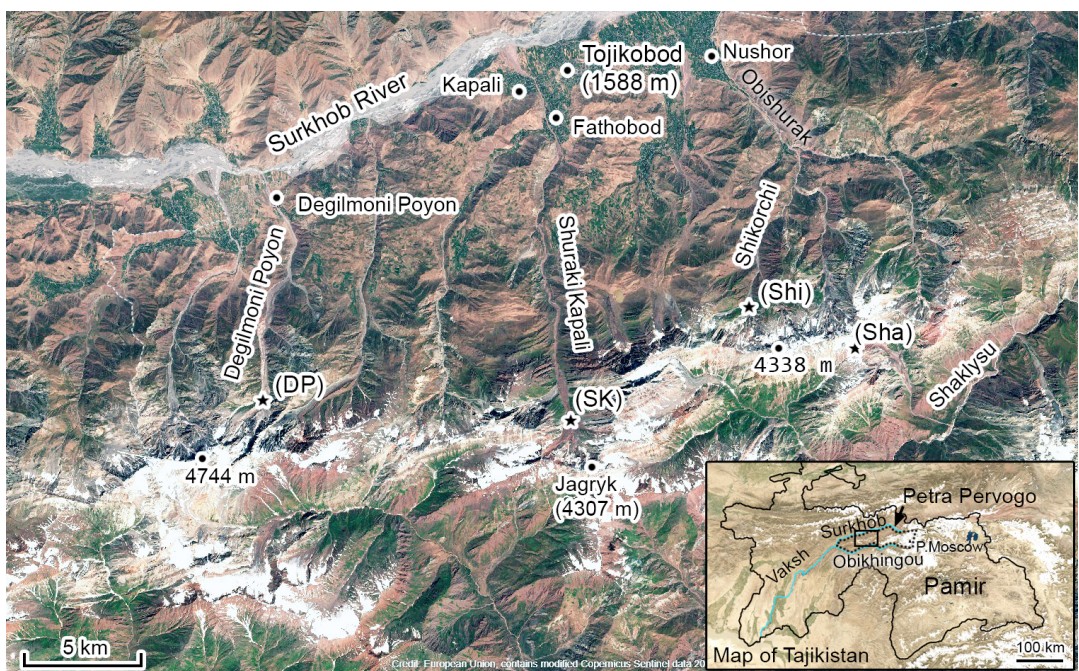

**Figure 1.** West Petra Pervogo Range. Symbols ⋆ indicates catchments where glacier detachments or ice avalanches occurred. Catchments are abbreviated by river names (DP, SK, Shi, Sha). Image contains modified Copernicus Sentinel-2 and MODIS data.

flows resulting from glacier detachments or rock/ice avalanches. Two detachments, which happened in 2016 and 2017, were mentioned in (Dokukin et al., 2019) and on twitter (Dokukin, 2018). A third detachment, which happened in 2019, was found during this study and independently by (Kääb, 2020).

To analyze recovery times of vegetation after large mass flows we compared the run out of the Kolka-Karmadon rock/ice avalanche in the Caucasus (Russia) with the run out of two largest detachments in the Petra Pervogo range.

## 2.1 Catchments in the Petra Pervogo range with large mass flows

In the catchment of the *Degilmoni Poyon* river (*DP* in Fig. 1) we identified a glacier detachment which occurred in 2019 (abbreviated as dp-19). It resulted in a mass flow which almost reached the village Degilmoni Poyon, located 9 km downstream.

The glacier detached between 2860 and 3360 meters altitude at about 38.988° N, 70.694° E.

In the catchment of the *Shuraki Kapali* river (*SK* in in Fig. 1), 13 km upstream of the village of Tojikobod (Tadzhikabad, 1737 inhabitants (Wikipedia, 2017), 1588 m a.s.l.), a series of detachments and ice avalanches occurred between 1973 and 2019, abbreviated as sk-YY. The nearby villages Kapali and Fathobod suffered from damage of infrastructure on 28 August 2016. The largest detachment in 2017 is abbreviated as sk-17. Ice masses detached between 3300 and 4000 m of altitude at

about 38.974° N, 70.844° E.





In a side valley of the *Shaklysu* river (*Sha*), two large ice-rock avalanches, very likely detachments occurred in 2006 and 2019. The 2019 avalanche traveled through the side valley and almost reached the Shaklysu river. The originated at 3800 m altitude at a small glacier at 39.012° N, 70.998° E.

In the catchment of the *Shikorchi* river (*Shi*), a series of large avalanches (2000–2017), most of them rock-ice avalanches, was identified, originating between 3000 and 4000 m altitude at (39.026° N, 70.933° E).

## 2.2 Geology

The north west of the Pamir system shows a particulary high density of surging glaciers (Goerlich et al., 2020), spatially correlated with the occurrence of soft and fine-grained sediments like limestone, claystone, sanstone, conglomerates, aleurolite, gypsum and marl (see geological map by Ibrohim et al.). The Western Petra Pervogo range is composed mainly of Cretaseous–Neogene sedimentary rocks. The catchments DP and SK are made up of redstone, aleurolite, claystone, conglomerates and limestone. Striking erosional features and thick glacial debris cover support the fact that soft lithologies are widespread in the Petra Pervogo Range. The Petra Pervogo range is located south of the Vakhsh thrust system and shallow earthquakes in the upper 15 km of the crust are frequent (Schurr et al., 2014).

## 2.3 Climate conditions

Two stations at Rasht/Garm (1316 m, 39.02°N, 70.37°E) and Lyairun (2008 m, 38.89°N, 70.93°E), located in the valleys of the Surkhob and Obikhingou river 40 km west and 12 km southeast of SK, indicate a mean annual precipitation of 700–1000 mm yr$^{-1}$ (Williams and Konovalov, 2008) and a mean annual air temperature (MAAT) of 10.7 °C and 7.1 °C resulting in a temperature-lapse rate of -0.52 °C per 100 m. A temperature increase of 0.42 °C over the last 40 years has been observed for the Pamir mountains, with an almost 1 °C increase in fall and winter (Finaev et al., 2016).

For the vegetation covered avalanche run out zones of about 2500 m we obtain a MAAT of +4.5 °C which is comparable to the a MAAT of +4.0 °C at the run-out zone of the Kolka-Karmadon rock-ice avalanche at 1800 m (Haeberli et al., 2004). At 3300 m the MAAT in the Petra Pervogo range is close to 0 °C and Obu et al. (2019) indicates isolated patches of permafrost for the region. Based on Sentinel-1 radar backscatter data we determined that snow melt at ∼ 4000 m starts around mid April every year, and melting temperatures last until October.

## 3 Data and methods

### 3.1 Satellite data for event identification

As almost no in situ information is available to us, the study is mainly based on remote sensing data. To identify and characterize detachments and ice avalanches we analyzed the entire Landsat archive (L1–L8), all available Sentinel-2 (S2) images, the archive of ASTER imagery, and selected Planet and reconnaissance Keyhole images. To characterize the events and to narrow





down the event date we compared consecutive images. We also compared images from different years but acquired at the same month of the year.

S2 and L8 imagery, available since November 2015 and April 2013 at a resolution of 10 m (ESA, 2015) and 15 m in the L8 panchromatic channel (USGS, 2013), were downloaded with the Sentinel EO Browser (Sinergise Ltd., 2020). For false-color visualization we used the NIR, red, and green band (B8, B4, B3) for S2 and the SWIR1, panchromatic, and red band (B6, B8, B4) for L8. L1–7 imagery since 1972 is available in the Google Earth Engine data catalogue and was processed using the Google Earth Engine. L7 (ETM+), available since 1999, has a panchromatic channel with 15 m resolution, therefore we used the SWIR2, panchromatic, and red band (B7, B8, B3) for visualization. L4 and L5 (TM) have 30 m resolution and L1–L3 (MSS) have 60 m resolution. L5 provided only 15 images between 1986 and 1989. No imagery of L3 and L4 were available from our study site. L2 provided 23 images (1975–1977) and L1 provided two images (1972/73). ASTER imagery at 15 m resolution, available since September 2000, were also processed with the Google Earth Engine. We used the NIR, red, and green band (B3N, B2, B1). To narrow down the occurrence of avalanches we also analyzed optical Planet imagery and Sentinel-1 (S1) radar imagery. As earliest images, we analyzed declassified panchromatic satellite images from the Keyhole missions (KH-3 and KH-7), dating back to 1961 and 1973, with resolutions between 7 and 10 m (USGS, 2008). The images were not orthorectified.

## 3.2 Avalanche characterization

To characterize detachments and ice avalanches, we measured the horizontal avalanche path length, the total fall height, the maximum height of the avalanche trim lines, and the maximum width of the avalanches with the Google Earth Pro measuring tool (elevation information based on the SRTM). To calculated the impacted area we mapped the total avalanche run out zone from satellite imagery with the software QGIS. The horizontal avalanche path length and fall height were measured from the avalanche crown or rupture line (Schweizer et al., 2003), to the lowest avalanche runout point. To assess the mobility of the mass movements, we calculated the mobility index *Fahrböschung*, also known as the *angle of reach* $\alpha$ resulting from the ratio of total fall height and horizontal runout distance, $\tan \alpha = H/L$ which corresponds to the average friction coefficient (Scheidegger, 1973). Snow avalanches have a typical Fahrböschung of 20–40° and debris flows between 10 and 20° (McClung and Gauer, 2018; Lied and Toppe, 1989; Prochaska et al., 2008).

## 3.3 Detachment volume estimation

To estimate the detached volume of the dp-19 event we obtained three pairs of World View stereo images (09 September 2018, 03 August 2019, and 10 April 2020; Neigh et al. (2013)) which, unfortunately, covered only the DP catchment. DEMs were generated using SETSM (Noh and Howat, 2017) and were coregistered following Nuth and Kääb (2011). Since the glacier detachment sk-17 showed a similar geometry and area as the dp-19 event, we assume the same ice thickness to estimated the volume of the detached area but estimated an additional error of 20%.



In the SK catchment, the DEM difference between the C-band SRTM (Farr et al., 2007) and the ALOS World DEM 3D (W3D) (Tadono et al., 2016) revealed a previously unknown event and the two DEMs were used to roughly estimate its volume. The event was constrained by satellite imagery to at least two events in 2003 and 2006 (abbreviated sk-03+06).

Volume uncertainties associated with all DEM differences were estimated follow the method described in (Miles et al.,
2018). To get a reliable estimate of uncertainty from World View images, we masked obvious clouds (large areas with a DEM difference beyond $\pm 130\,\mathrm{m}$) before performing the uncertainty assessment.

### 3.4   Glacier velocity prior to detachment

To identify detachment-precursors like increased sliding or strong crevassing on the glaciers before the two largest detachments (sk-17, dp-19) we used high resolution S2 and L8 imagery. Velocities immediately before detachment were determined by
manual tracking of surface features and by measuring the width of the opening rupture line.

### 3.5   Surge history of detached glaciers

To study the surge-history of the detached glaciers we used DEMs from SRTM, TanDEM-X (TDM), the W3D, and World View (WV) stereo imagery (Farr et al., 2007; Krieger et al., 2007; Tadono et al., 2016; Neigh et al., 2013; Noh and Howat, 2017). We analyzed six interferometric TDM pairs acquired between 03 May 2011 and 05 September 2014, and processed
DEMs as outlined in (Leinss and Hajnsek, 2018) to calculate DEM differences.

### 3.6   Meteorological and seismic data

To analyze climatic influences we used data from the two meteorological station Garm and Lyairun available from 1961–1990 and ERA-Land reanalysis data from 1981–2019 obtained for the coordinate 70.90°E, 38.95°N (3470 m), 6 km south east the SK catchment. To obtain homogeneous temperatures time series we calculated the mean difference between the Lyairun and
ERA-Land temperature and shifted the data of the Lyairun station by +7.6 °C, in agreement with a lapse rate of 0.52 °C per 100 m. ERA-Land precipitation required a scaling factor of 16.7 to match data from the Lyairun station.

To assess earth quakes as triggering factors, we used data of seismic events which occurred within a range of about 100 km around the Petra Pervogo range. The data was provided by the USGS via the IRIS Data Management Center.

### 3.7   DEM differences for detection of surging glaciers

To put the detachments and mass movements into a regional context, we mapped surging glaciers in the entire Pamir mountains from 2000 to 2011 by differencing the C-band SRTM and the optical W3D, horizontally aligned following Nuth and Kääb (2011). We analyzed DEMs from 37–39 North and 67–75 East. The SRTM DEM was acquired in February 2000 and is available at 1 arcsec resolution (30 m) from the USGS. An absolute vertical accuracy of 6 m is given in (Farr et al., 2007) but the C-Band radar can penetrate up to 10m into dry snow and firn (Rignot et al., 2001). Imagery for the W3D was acquired
between 2006 and 2011 (main acquisition phase between March 2008 and March 2011 in the Petra Pervogo range) therefore




**Table 1.** Characteristics of glacier detachments and other events. Empty spaces indicate unknown quantities, dashes indicate quantities without meaning. Surge-like instabilities were observed several years before the sk-17 and dp-19 events but not immediately before detachment. *The sk-16b event transformed into a debris flow, possibly after entrainment of material of the sk-16a event. Due to lack of data we could not determine which fraction of the total length of 19.1 km belongs to the initial ice avalanche and and which to the subsequent mud flow.

| abbreviation | sk-73 | sk-03+06 | sk-04 | sk-16a | sk-16b | sk-17 | sk-19 | dp-19 | sha-06 | sha-19 | shi-01 | shi-09 | shi-17 |
|---|---|---|---|---|---|---|---|---|---|---|---|---|---|
| **Release area** | | | | | | | | | | | | | |
| Avalanche type | detach(?) | detach | detach | ice | detach(?) | detach | ice/rock | detach | detach(?) | detach | rock/ice | | |
| Area ($10^3$ m$^2$) | 220 | 243 | 75 | 95 | 160 | 250 | 57 | 244 | 41 | 93 | | | |
| Volume ($10^6$ m$^3$) | | $3.2 \pm 0.3$ | | | | $8.8 \pm 2.7$ | | $8.6 \pm 0.9$ | | | - | - | - |
| Horiz. length (m) | 1050 | 940 | 685 | 520 | 1010 | 1020 | 533 | 1170 | 365 | 700 | - | - | - |
| Slope (°) | 18.5 | 22.1 | 19.8 | 24.4 | 22.3 | 15.6 | 25.7 | 21.9 | 24.9 | 24.2 | 37.6 | 26.7 | 34.3 |
| Surge observed | | | no | no | no | (yes) | no | (yes) | no | no | - | - | - |
| **Avalanche run out** | | | | | | | | | | | | | |
| Size (km$^2$) | 0.71 | 1.17,0.50 | 0.42 | 1.01 | 2.70 | 1.91 | 1.68 | 1.83 | 0.27 | 0.69 | 0.53 | 0.69 | 0.78 |
| Horiz. path length (km) | 3.34 | 7.3, 3.4 | 2.9 | 5.4 | 19.1* | 8.5 | 9.0 | 6.7 | 1.9 | 4.7 | 5.2 | 5.3 | 4.5 |
| Height difference (m) | 830 | 1520, 850 | 790 | 1200 | 2590 | 1520 | 1930 | 1525 | 660 | 1320 | 1680 | 1680 | 1600 |
| Angle of reach $\alpha$ (°) | 14.0 | 11.8, 14.0 | 15.2 | 12.4 | 7.7 | 10.1 | 12.1 | 12.7 | 19.1 | 15.8 | 17.8 | 17.6 | 19.6 |
| Max width (m) | 220 | 230 | 345 | 350 | 650 | 535 | 600 | 420 | 250 | 290 | 300 | 130 | 270 |
| Max trim line height (m) | | | | 40 | 71 | 53 | 53 | 168 | ? | 92 | 10 | 20 | 30 |
| Figure reference | A2 | A3a,b | A4a | A3c | A4b | 6 | A4c | 4 | 8d | 8b | A5a | A5c | A5b |

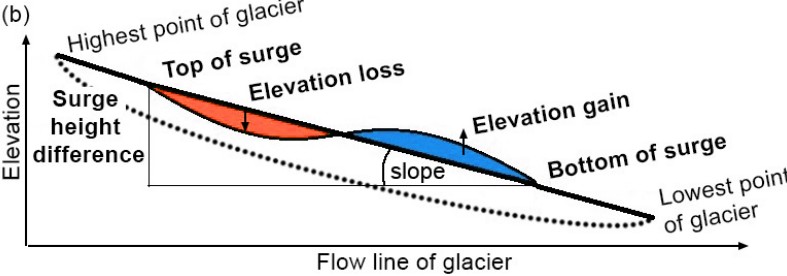

**Figure 2.** Sketch of measured parameters for surging glaciers.





**Table 2.** Satellite imagery used to limit the date of occurrence of the events. Date are given according to ISO-8601 (YYYY-MM-DD). Event types are abbreviated as d (detachment), i (ice avalanche), i/r (ice-rock avalanche), r/i (rock-ice avalanche). Referred figures show images with best visibility of the events; shown images do not necessarily agree with the images used to limit the date of occurrence.

| Event | type | pre-event image | post-event image | shown in |
|-------|------|-----------------|------------------|----------|
| dp-19 | d | 2019-08-02, S2 | 2019-08-03, L8 | Fig. 4 |
| sk-73 | d | ∼1973, KH | 1973-08-03, KH | Fig. A2 |
| sk-03 | d | 2003-08-24, L7 | 2003-09-25, Aster | Fig. A3a |
| sk-04 | d | 2004-09-02, L7 | 2004-09-18, Aster | Fig. A4a |
| sk-06 | d(?) | 2006-08-23, L7 | 2006-09-01, L7 | Fig. A3b |
| sk-16a | i | 2016-07-14, S1 | 2016-07-25, L8 | Fig. A3c |
| sk-16b | d(?) | 2016-08-27, L7 | 2016-08-31, S1 | Fig. A4b |
| sk-17 | d | 2017-07-10, Planet(?) | 2017-07-11, S2 | Fig. 6 |
| sk-19 | i/r | 2019-06-21, S2 | 2019-06-23, S1 | Fig. A4c |
| sha-06 | d | 2006-08-16, L7 | 2006-09-01, L7 | Fig. 8d |
| sha-19 | d | 2019-07-06, Planet(?) | 2019-07-08, L8 | Fig. 8b |
| shi-01 | r/i | 2001-03-11, L7 | 2001-03-18, L7 | Fig. A5a |
| shi-09 | r/i | 2009-04-09, L7 | 2009-05-11, L7 | Fig. A5c |
| shi-17 | r/i | 2017-06-01, S2 | 2017-06-08, S2 | Fig. A5b |

no precise time stamp is available. The W3D is commercially available at 5 m resolution and has a vertical accuracy is of 5 m (Tadono et al., 2016). Here we used the freely available 30 m version.

We considered a glacier being in its active surge phase when the glacier showed a surface height increase of more than 10 m over the glacier tongue accompanied by surface lowering further upstream. We consider glaciers being in a quiescent 165 surge phase when surface lowering over the glacier tongue exceeded 10 m and a significant surface height increase was visible upstream, in a possible reservoir area. To determine the slope of the surging part of a glacier we measured the horizontal length and the elevation difference of the surge-like elevation change pattern as illustrated in Fig. 2.

### 3.8 NDVI for mass flow recognition and vegetation recovery analysis

The older, available imagery showed gaps of a few years in which detachments could have happened without being noticed. 170 However, mass flows with long runouts may remove or bury vegetation which can take years to recover. To assess recovery times we analyzed time series of the $NDVI = (NIR - Red)(NIR + Red)$ from the band combinations (B5, B4) and (B8, B4) for LS8 and S2, respectively, of the two recent detachments, dp-19, and sk-17. We compare the results with vegetation recovery in the run out of the Kolka-Karmadon glacier detachment.




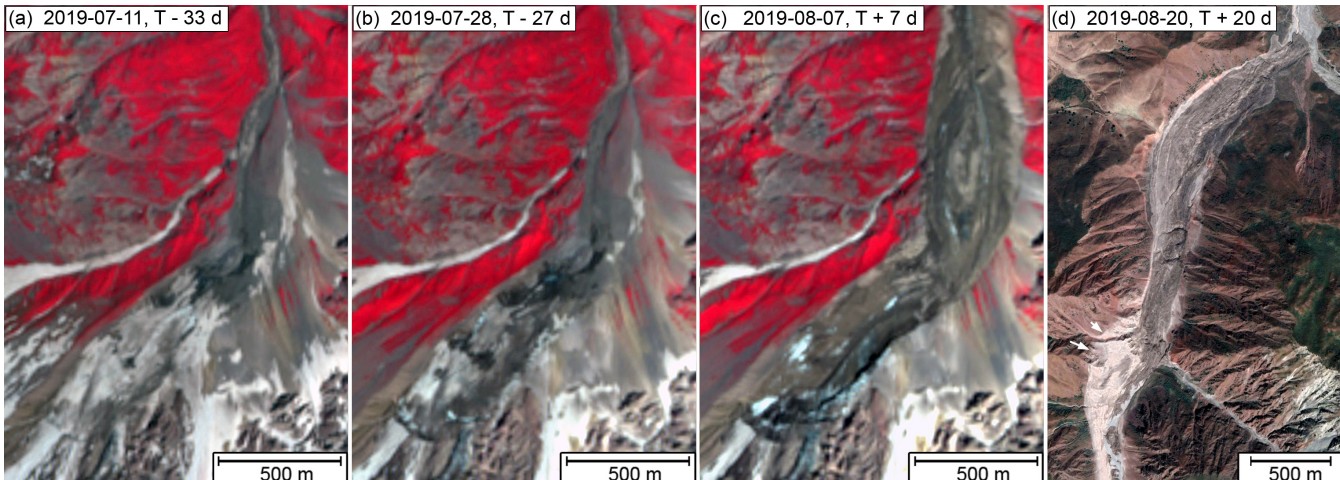

**Figure 3.** S2 false color imagery capturing the evolution of the detachment dp-19. From (a) to (b) increased crevassing is visible. (c) shows the detached glacier, (d) the run out zone of the resulting avalanche. (a-c) Copernicus Sentinel data (2019). (d) ©Google, Maxar Technologies.

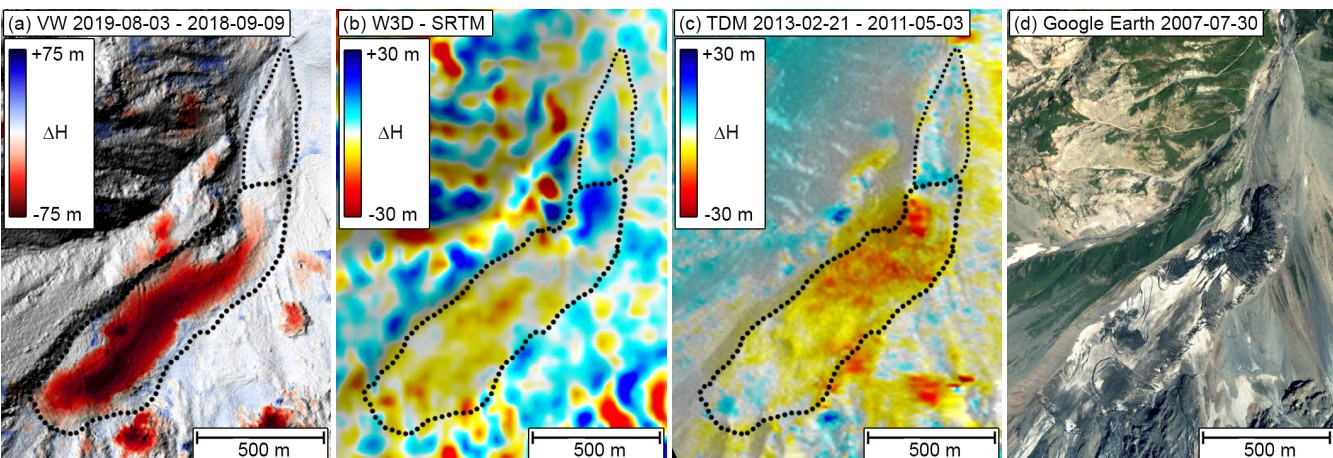

**Figure 4.** DEM differences of the detachment dp-19. (a): WorldView elevation differences from before and after the event reveal a detached volume of $8.6 \times 10^6 \, \mathrm{m}^3$; 2020, DigitalGlobe; NextView License. (b, c) DEM differences indicate a surge-like elevation change pattern after 2000 which continued at least until 2013. (d) In 2007 strong crevassing resemble surge-like dynamics. ©Google, Maxar Technologies.

## 4 Results

Table 1 summarizes the characteristics of the detected events and Table 2 lists satellite images used to narrow down their date of occurrence. The following sections describe the events in detail.





### 4.1 Degilmoni Poyon (DP) glacier detachment 2019

In the DP catchment we identified a valley glacier which detached between 02.08.2019 and 03.08.2019 (Fig. 3). The glacier is listed in the GLIMS data base with the ID G070689E38981N (Raup et al., 2007) and its outline comprises a headwall with 180 hanging ice and a glacier which feeds from the headwall. The detachment, abbreviated as dp-19, involved essentially the entire glacier below the headwall.

From the difference of two WorldView DEMs from 2018 and 2019, shown in Fig. 4a, we determined a detached area of approximately $244 \times 10^3 \, \mathrm{m}^2$ and a detached volume of $8.59 \pm 0.88 \times 10^6 \, \mathrm{m}^3$. A cloud obscured a small part of the detachment area in the 2019 image, but the DEM difference between a 2020 and the 2018 DEM indicated that only a minimal part of 185 the detachment area is obscured. The post-detachment glacier bed showed a nearly triangular cross section, with a maximum erosion depth of $91 \, \mathrm{m}$ (mean depth: $35 \, \mathrm{m}$). The detached mass travelled $6.7 \, \mathrm{km}$ down the valley, with an elevation loss of $1525 \, \mathrm{m}$, resulting in a angle of reach of $\alpha = 12.8°$. After traveling $4.3 \, \mathrm{km}$ down valley, the avalanche trim line reached over $150 \, \mathrm{m}$ above the valley in a curve indicating a very high velocity (arrows in Fig. 3d). The avalanche stopped $2.4 \, \mathrm{km}$ later, reaching it's end approximately $2.6 \, \mathrm{km}$ before the village Degilmoni Poyon.

DEM differences prior to the detachment (Fig. 4b,c) indicate that the glacier surged between 2000 (SRTM) and 2006–2011 (W3D). L7 imagery indicates an advance of about $100 \, \mathrm{m}$ between 1999 and 2003, followed by quiescence until until 2006. In a Google Earth Pro image from 30.07.2007 the glacier appears heavily crevassed (Fig. 4d), indicating an active surge phase. It advanced again by about $40 \, \mathrm{m}$ in total until 2013. TanDEM-X data from 03 May 2011 and 21 February 2013 (Fig. 4c) indicate an elevation loss of about 10–15 meters over the later detached area. In the following years, satellite imagery suggests healing 195 of the crevasses and no detachment or avalanche could be found between 2007 and 2019. Between 2013 and 2018 satellite imagery, and DEM differencing indicates melt and retreat of the previously advanced tongue. Only about three weeks prior detachment, around 11 July 2019, the Bergschrund had started widening by $1 \, \mathrm{m} \, \mathrm{d}^{-1}$ and we observed increased sliding leading and enhanced crevassing around the detached area.

Erosion patterns and missing vegetation matching surprisingly well the avalanche patterns shown in Fig. 3d are visible in 200 Google Earth imagery and already in an KH image from 1961 but we could not find any confirmation of an earlier avalanche.

### 4.2 Shuraki Kapali (SK)

In the upper catchment of the Shuraki Kapali river, for which the GLIMS database lists five small glaciers, we identified a series of glacier detachment and several other mass flows which could not clearly be identified. Fig. 5 shows the locations of the released masses (in color) and GLIMS glacier outlines in magenta. About three kilometers downstream of the headwall, a 205 glacier which has surged in 2010/11 enters the catchment area from the west.

#### 4.2.1 Shuraki Kapali glacier detachment 2017

A large-volume glacier detachment was reported in this catchment by Dokukin et al. (2019). Between 10 and 11 July 2017, almost the entire valley glacier (GLIMS ID G070852E38974N) with an area of about $250 \times 10^3 \, \mathrm{m}^2$ detached (Fig. 6). Based




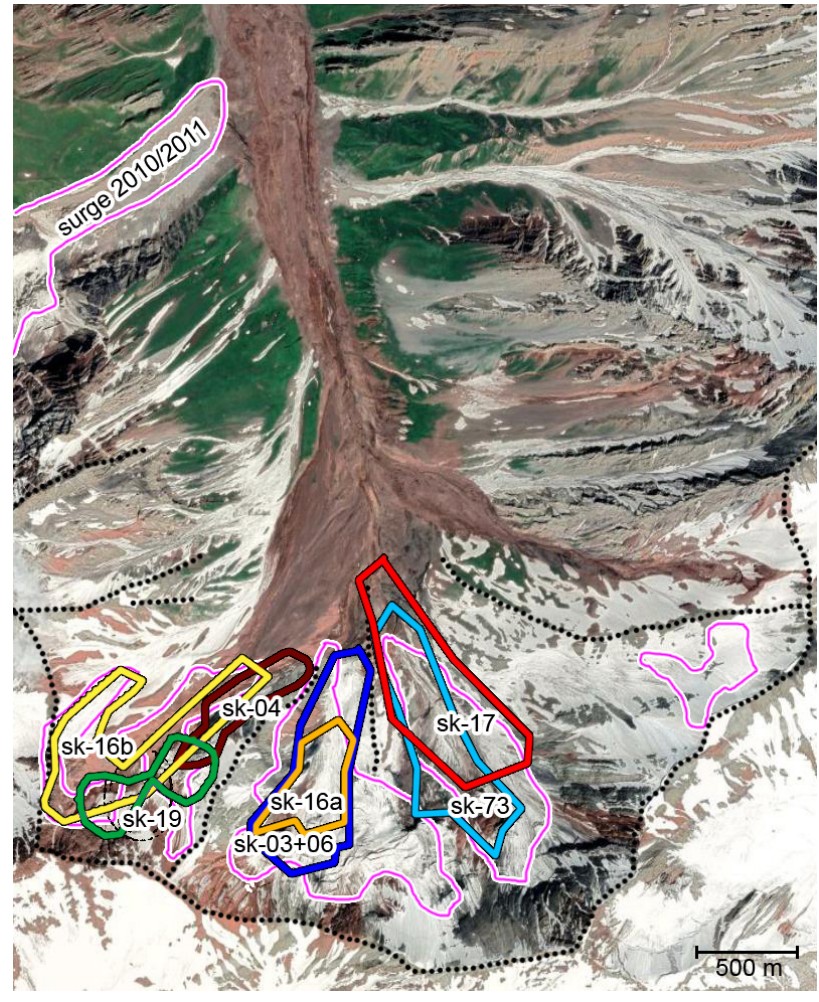

**Figure 5.** Shuraki Kapali catchment area and location released ice masses (color). Polygons in magenta correspond to the outlines of the GLIMS database, black dotted lines mark terrain ridges. Image from 19 July 2019 ©Google, Maxar Technologies.

on the mean depth of 35 m of dp-19, we estimated a volume of $8.8 \times 10^6 \, \text{m}^3$. The detached mass lost 1520 m in elevation while
travelling 8.5 km down the valley, which corresponds to an angle of reach of $\alpha = 10.1°$.

Figs. 6(a-c) show the evolution of the glacier prior to the detachment. Some crevassing, unusual compared to previous years, becomes visible 60 days before the detachment, and heavy crevassing indicative of enhanced sliding is visible 20 days before the detachment. Manual tracking of surface features in an S2 image pair from 21 and 28 June 2017 indicates a sliding velocity of about $3 \, \text{md}^{-1}$. Two weeks later, the glacier detached.

DEM differences prior to the detachment indicate a surge-like elevation change between 2000 and 2006–2011 (Fig. 7a) which continued until 2011 (TDM). However, prior to the detachment the glacier's surface elevation seems nearly stagnant,




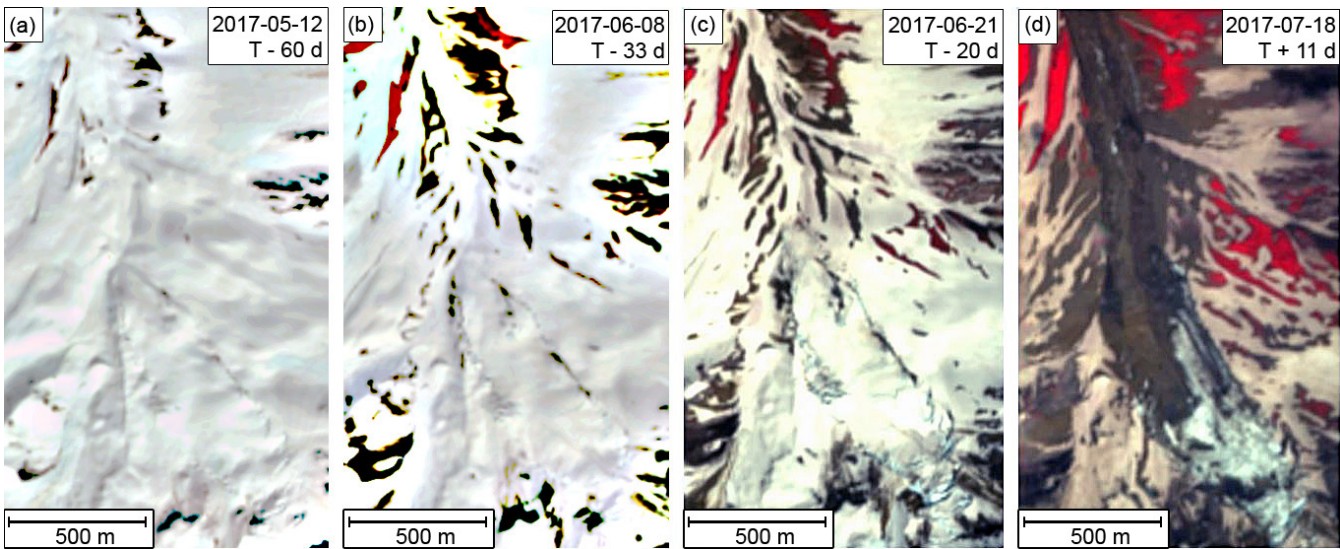

**Figure 6.** S2 false color imagery capturing the evolution of the detachment sk-17. Copernicus Sentinel data (2017).

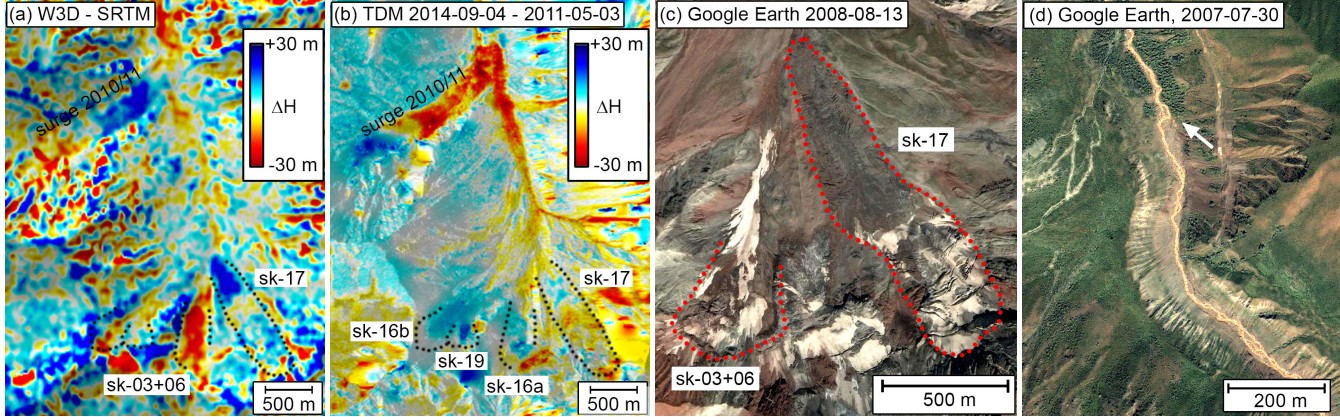

**Figure 7.** Shurali Kapali catchment. (a) the W3D-SRTM DEM difference shows a clear height loss due to detachment sk-03+06 (red) and a surge-like elevation gain at the tongue of sk-17 (blue). (b) the DEM difference between 2011 and 2014 shows hardly any surging before the sk-17 event; some ice is moving into the sk-03+06 detachment area and a strong elevation loss can be seen at the confluence of the 2010/11 surge and the valley floor, likely due to melt of ice and deposits. (c) shows the possible rupture line of sk-03 or sk-06; (d) suspected end of the sk-03 avalanche indicated by existing tall vegetation at the valley floor (arrow). Imagery (c) and (d) ©Google, Maxar Technologies.

and TDM DEM differences show hardly any advance between 2011–2014, Fig. 7b, and TDM and L8 imagery don't show any retreat either.

For the same glacier, we found evidence for an earlier detachment, sk-73, in a Keyhole (KH) image from 03 August 1973, shown in Fig. A2b. From the run-out distance of 3.3 km and the estimated height difference we obtain an angle of reach of around 14°. This is considerably lower than usual for snow avalanches (20–40°) (McClung and Gauer, 2018) and resembles




a similar angle of reach as obtained for the two detachments dp-19 and sk-17 (Table 1). Avalanche-like deposit pattern in an earlier KH image from 30 August 1961, Fig. A2a, and widening of the valley until 1973 indicate that large mass flows have occurred already before and have filled the valley before the 1973 ice avalanche occurred.

### 4.2.2 SK events in 2003, 2006, 2016

In the height difference between the SRTM and the W3D, red in Fig. 7a, we found a height loss of up to 40 m (13 m on average), indicated by "sk-03+06". The same area is shown in in Fig. 5 (dark blue outline) and is located on a glacier listed with the ID G070846E38972N in the GLIMS data base. From satellite imagery and DEM differences we estimate an approximate area of $243 \times 10^3 \, \text{m}^2$ and a volume loss of $3.2 \times 10^6 \, \text{m}^3$. A possible rupture line is visible on a Google Earth image from 13 August 2008 (Fig. 7c).

An inspection of satellite imagery revealed that at least two ice avalanches are responsible for the visible height loss. One occurred between 24 August and 25 September 2003 (sk-03) with a runout distance of about 7.3 km, Fig. A3a, while loosing 1520 meters in altitude, corresponding to an angle of reach of 11.8°. The run-out is clearly visible in Aster imagery (inset in Fig. A3a) and matches with missing vegetation on the valley floor in Fig. 7d. The comparison of two L7 image from August 2003 and 2004 reveals the detached area (white rectangles in Fig. A3a). The other event occurred between 23 August and 01 September 2006 (sk-06) and had a runout of 3.4 km, Fig. A3b. Though the avalanche is clearly visible on L7 imagery, we could not identify the exact release area and can therefore not classify it unambiguously as detachment.

In 2016 another ice avalanche originated from the same area (sk-16a, orange in Fig. 5), also mentioned by Dokukin et al. (2019). The avalanche occurred between 14 and 25 July 2016, Fig. A3c, travelled 5.6 km over a height loss of 1200 m, corresponding to an angle of reach of 12.4°. TDM imagery and DEM differences indicate that the sk-03+06 area has partially filled up with ice.

### 4.3 SK events in 2004, 2016, 2019

In the western part of the SK catchment area the GLIMS data base lists two small glaciers from which three ice avalanches originated. Extensive debris cover on the glacier made a precise delineation of the detached areas difficult.

Between 02 and 18 September 2004 the lower parts of a glacier with the GLIMS ID G070839E38975N detached (sk-04) and resulted in an ice avalanche with an approximate run out distance of 2.9 km. The detachment zone and the avalanche are visible in Fig. A4a. Additional ice fell off the detachment zone a few days later at the location indicated by the arrow in the inset of Fig. A4a.

Local media report a mud-flow which occurred on 28 August 2016 as a result of glacier break off (Tajik telegraph agency, 2016; Radio Ozodi, 2016). Based on satellite imagery we determined a glacier area of $160 \times 10^3 \, \text{m}^2$, indicated as sk-16b in Fig. 5, which detached, corresponding to the major part of the glacier with the GLIMS ID G070835E38972N. The detachment scarp and the avalanche trim line are indicated by an arrow and a white dotted line in Fig. A4b. The avalanche reached or run over the deposits of the sk-16a event and transformed into a debris-flow of a remarkable runout distance of 19.1 km (measured from the detachment scarp) resulting in a very low angle of reach of only 7.7°. The avalanche passed villages of Fathobod and





Kapali, Fig. 1, where ten buildings and a bridge were damaged or destroyed and several cattle were swept away. The mud-flow reached the Surkhob River at 1507 m of altitude (inset in Fig. A4b), still containing pieces of ice according to photographs in media, and blocked temporarily the Shuraki Kapali river (Radio Ozodi, 2016). The total path length of 19.1 km is the combined length of the glacier detachment and the debris flow.

Between 21 and 23 June 2019 an ice avalanche (sk-19) was released from the same area as sk-16b, followed by one or two minor avalanches between 26 June and 01 July 2019 as shown in Fig. A4c. The run out distance of the main avalanche is approximately 9 km with an angle of reach of 12.1°. In total an area of approximately $56.8 \times 10^3$ m$^2$ detached, however, neither a clear rupture line could be identified nor a glacier was visible on this location because of very strong sediment coverage. Still, the location is listed as a glacier with the ID G070839E38975N in the GLIMS data base.

### 4.3.1 Detachment 2019 near the Shaklysu river

In a side-valley of the Shaklysu river a large avalanche occurred between 06 and 08 July 2019, originating from the upper reaches of a very small glacier with the GLIMS ID G070995E39014N at 3810 m. Exposed rocks at the former location of the glacier in Google Earth imagery indicate that the entire glacier has detached (Fig. 8). The resulting avalanche travelled 4.7 km over a vertical distance of 1320 m with an angle of reach of 15.8° and almost reached the Shaklysu river. No volume was estimated for this event. A maximum trim line height of 92 m indicates a high avalanche velocity. Satellite imagery indicate that the glacier was not existent in 2013 but build up mass until detachment in 2019.

For the same glacier, satellite imagery indicate an that an earlier detachment has likely occurred between 16 August and 01 September 2006 (insets in Fig. 8).

### 4.4 Avalanches in the Shikorchi (Shi) catchment

In the catchment of the Shikorchi river, we identified a series of large mass flows which travelled over steep glaciers but we could not determine how much ice was involved.

In the eastern part of the catchment, likely a rock fall (shi-01), occurred between 11 and 18 March 2001. It originated at 4000 m at the ridge of the catchment and above a glacier with the ID G070941E39016N, run over another glacier with the ID G070934E39019N, and traveled in total 5.2 km over an elevation difference of 1680 m, Fig. A5a. At the same location two avalanches (shi-17) occurred between 01, 08 and 21 June 2017, possibly triggered by rock fall. They run over the two glaciers, and the longest of these avalanche had a run out distance of 4.5 km over an elevation distance of 1600 m, Fig. A5b.

In the western part of the catchment, we identified a mass flow which occurred between 09 April 2009 and 11 May 2009, originated above the glacier with the ID G070926E39021N, traveling 5.3 km over 1680 m elevation, Fig. A5c. For none of these events we found any detached glacier and the origin of the avalanches rather indicates major rock fall events. This is supported by the relatively steep slope (26–38°) and high angle of reach (17.6–19.6°) as listed in Table 1.

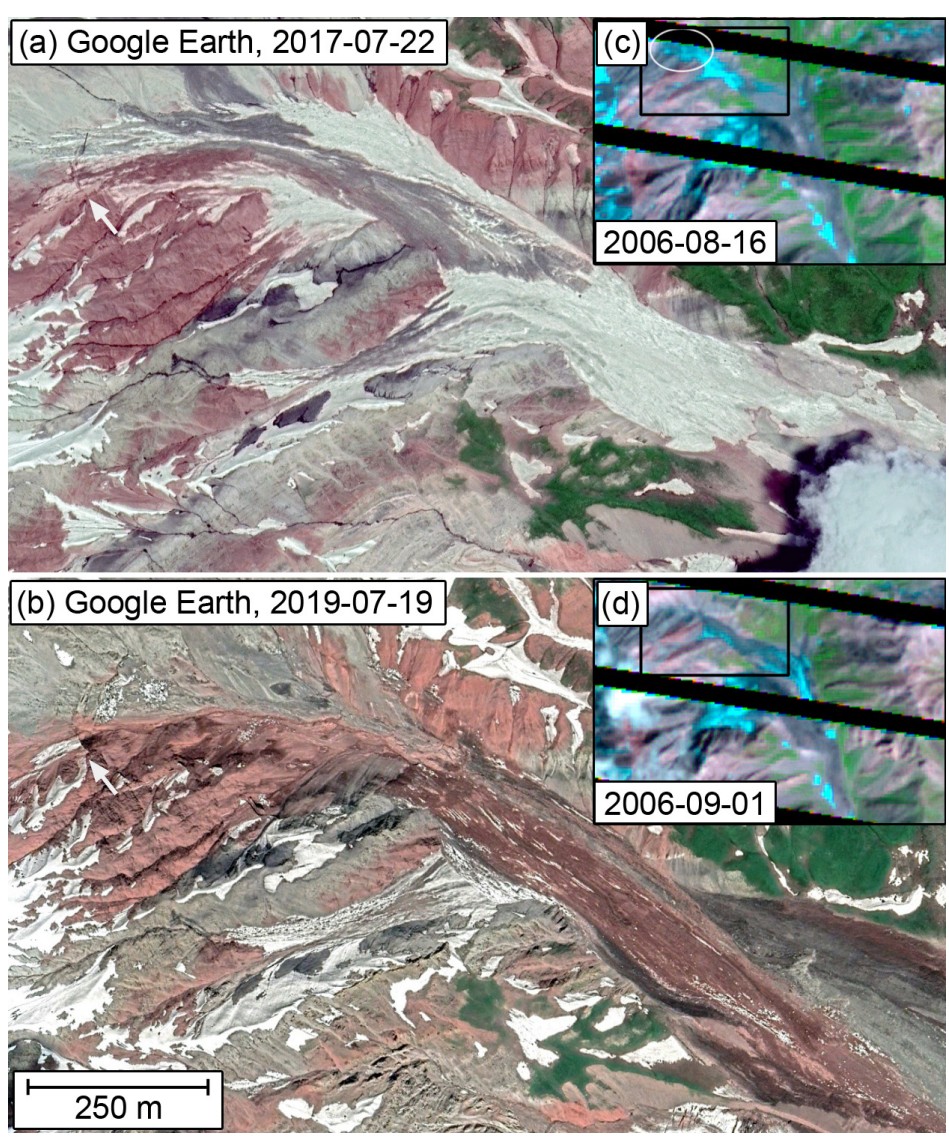

**Figure 8.** Imagery from before (a) and after (b) the small glacier detachment sha-19 in a side-valley of the Shaklysu river. A white arrow indicates the detachment scarp. (c, d) at the same place (white circle) an earlier detachment (sha-06) has very likely happened. The black box in the inset corresponds to the outline of the main images. (a, b) ©Google, Maxar Technologies. (c, d) Landsat-7 imagery courtesy of the U.S. Geological Survey.

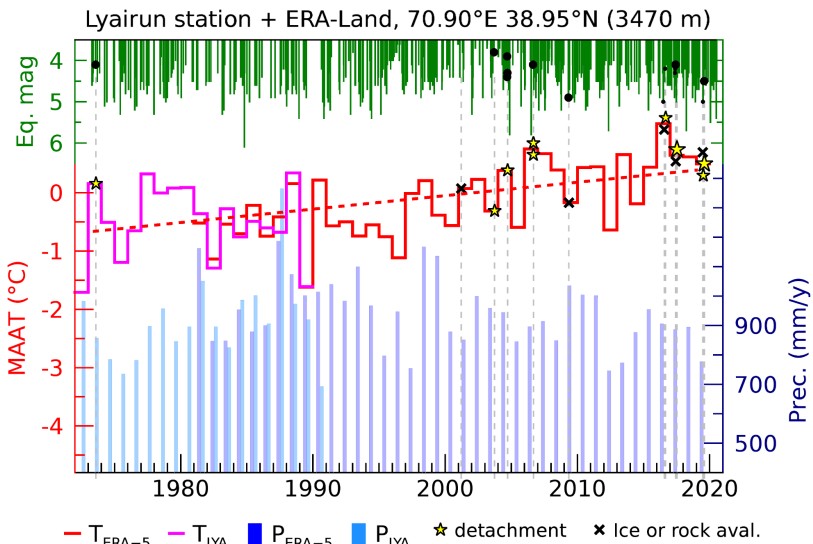

**Figure 9.** Mean annual air temperature (cyan) at the Lyairun station at 2008 m a.s.l. (Williams and Konovalov, 2008) shifted by -7.6°C to match the ERA-5 Land reanalysis data (Copernicus Climate Change Service (C3S), 2019) obtained for 3470 m a.s.l. (red). Detachments and other events are indicated by ⋆ and + symbols and are vertically distributed when more than one event occurred in the same year. Seismic events (green) with mag >3.5 are frequent within a radius of 100 km of the Petra Pervogo range: black bullets indicates earth quakes which occurred between the pre- and post-event image (Table 2) and black dots are earth quakes which occurred up to 14 days before the pre-event image. Figure contains modified Copernicus Climate Change Service Information (2020); Earth quake data from USGS via IRIS Data Services.

## 4.5 Meteorology and seismic activity

Almost all detachments (eight out of nine) and 12 out of all 14 events occurred in years where the mean annual air temperature (MAAT) was above the long-term trend (Fig. 9). Only the sk-06 detachment and the shi-09 rockfall event occurred in years with a MAAT below the trend. We interpret this in the sense that temperature has a very strong impact on the occurrence of glacier detachments. No correlation to precipitation was found.

The magnitude of all earth quakes which occurred within a radius of 100 km of the SK catchment are shown in Fig. 9 as green bars. Earth quakes which occurred between the pre-event and post-event satellite image according to Table 2 are shown as black bullets; black dots indicate earth quakes which occurred up to 14 days before the pre-event image.

For earthquakes with a magnitude above 5.0 we observed no detachment or major rockfall event. On 06 July 2006 an mag 5.8 earth quake occurred 90 km southeast of the SK catchment area but 1.5 months before the sk-06 event. The largest earth quake (mag 4.9, 29 April 2009) which occurred within the possible time-period April–May 2009 of the rock avalanche shi-09 happened 67 km east of the catchment in a year with below-trend temperatures. The second largest earth quake (mag 4.5, 03 August 2019) which occurred within the time period of the dp-19 detachment happened 55 km west of the dp-19 detachment. Other earth quakes were below mag 4.5. Although several earth quakes happened in temporal proximity of the detachment or

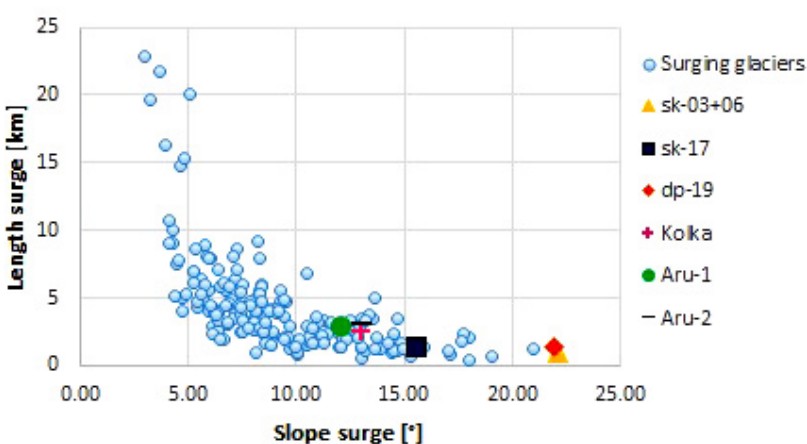

**Figure 10.** Horizontal length over slope of the surging part of glacier in the Pamir Mountains and surface slope of the detached parts of glacier detachments.

rock fall events, there were a large number of earth quakes which are not related to any event, especially the stronger ones.
Therefore we conclude that is is very unlikely that the detachment events were triggered by seismic activity.

### 4.6 Comparison with surging glaciers in the Pamir

In total, we identified 237 glaciers in the entire Pamir mountains which were either in a surge- or in a quiescence phase. Of these 188 showed both an elevation increase at the terminus and a decrease further up, 32 glaciers showed only an elevation increase at the terminus and 17 seemed to be in a quiescent phase.
The comparison in Fig. 10 of the slope and length of all surging glaciers with the detached glaciers of the the largest events, dp-19, sk-17, and sk-03+06, and in addition with the Aru- and Kolka-Karmadon detachments, shows that glacier detachments occur predominantly for short but steep glaciers, at least when compared to glaciers which showed a surge-like instability in the past.

### 4.7 Retroactive avalanche detection using NDVI

The largest avalanches in this study, sk16-b, sk-17, sk-19, and dp-19, were identified in satellite imagery by destruction of vegetation in the associated valleys. Unfortunately, most other avalanches travelled in already eroded valleys, therefore it was difficult to detect them by means of vegetation change only and the panchromatic channels of L7 and L8 provided more spatial details than the NDVI. Nevertheless, an analysis of time-series of the NDVI evolution in the DP- and SK-catchment, Fig. 11, shows that it within two years between events the vegetation hardly recovers. In the run out zone of the Kolka-Karmadon
detachment, where a suitable long satellite time series exist and where no repeated avalanches occurred, vegetation recovery to pre-detachment NDVI values takes around 10 years (Fig. A1). Because of similar climatic conditions, we conclude that the




chance of missing long run outs of mass flows that reach vegetated areas is very low. Therefore we are confident that we did not miss any major events.

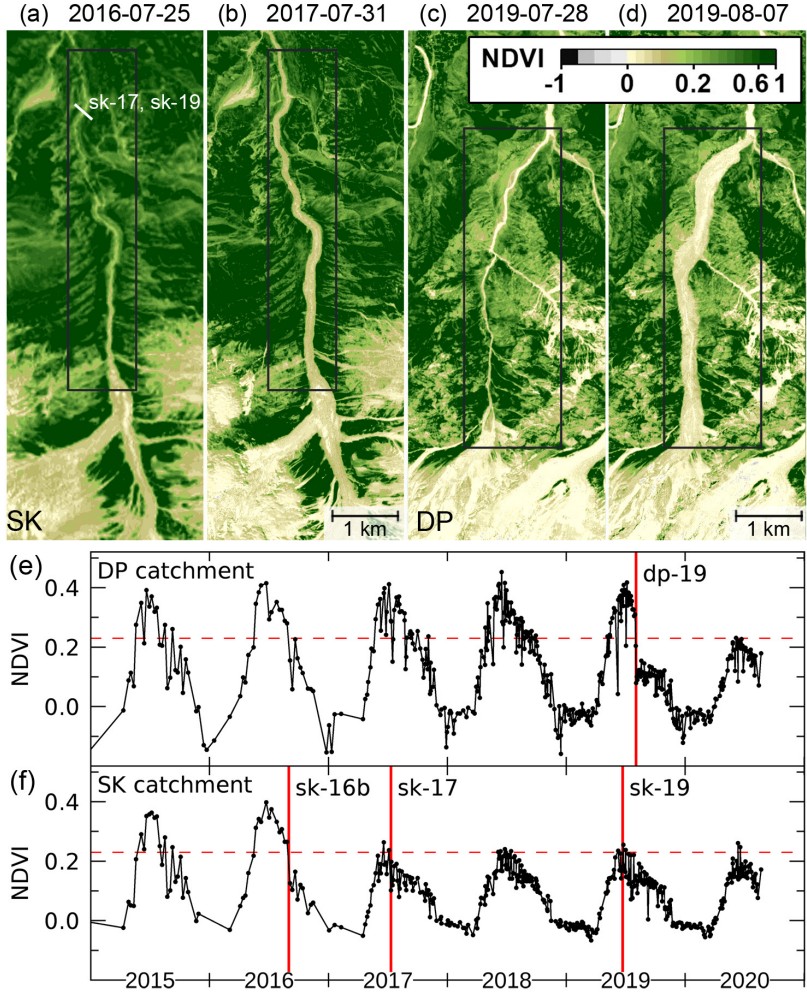

**Figure 11.** (a) NDVI before the sk-16b detachment. The white bar indicate the end of the run out of the sk-17 and sk-19 event, the sk-16b mud flow traveled further, (b) NDVI after the sk-17 detachment, (c, d) NDVI before and after the dp-19 detachment. The time series of the mean NDVI obtained from L8 and S2 over the eroded area in the black box show that vegetation does hardly recover within two years. Copernicus Sentinel data (2020) and Landsat-8 image courtesy of the U.S. Geological Survey.

## 5   Discussion

The numerous recent discoveries of glacier detachments around the world (Kääb et al., 2018; Gilbert et al., 2018; Falaschi et al., 2019; Paul, 2019; Jacquemart et al., 2020) have raised important questions about the conditions and triggers leading a



glacier to detach. Our analysis of the 46-year of satellite record over the Petra Pervogo range has revealed a cluster of such events in a small geographical area that provides additional understanding of these catastrophic events, in particular with regard to the link between surging glaciers and glacier detachments, and the influence of climate change and seismic activity.

## 5.1 Detachment detection

Analyzing the entire satellite record is frequently the only way to assess the occurrence of past large mass flow events in a given geographical area (Coe et al., 2018; Bessette-Kirton and Coe, 2020). This approach is not fool proof, since clouds and shadows may have obstructed detection of certain events, but we compared multiple consecutive images and in addition images acquired in the same month of consecutive years, therefore we are reasonably certain that we did not miss any large

events, especially in more recent years. While the traces left by smaller events easily disappear against the background of loose sediment and hillslopes free of vegetation, large events that reach vegetated areas leave distinct traces that can be detected for several years. Our analysis of vegetation recovery at Kolka-Karmadon (approximately 10 years) and the fact that we discovered sk-17 and dp-19 in this fashion demonstrate how the NDVI and the vegetation sensitive NIR channel are good means to detect long-runout events in remote sensing imagery, even years after they happened. Closer to the source, where

vegetation in strongly eroded valleys is missing, the moisture sensitive channels SWIR1 and SWIR2 of Landsat -7 and -8 allow for detection of sediment-covered ice, at least several weeks after detachment; in addition, they allow for separation of snow and clouds. Lastly, the low resolution of 30 and 60 m of Landsat 1–5, which lack a higher resolution panchromatic channel, could impede the detection of some events. To complement the drawbacks of all optical methods (especially in areas with poor color contrast), and in particular for the detection of more recent events, differencing high resolution DEMs is undoubtedly the

most reliable way to detect drastic changes in glaciated catchments. We found that weather-insensitive radar imagery is helpful to detect abrupt changes but the bright backscatter signatures of avalanches disappears quickly within a few days. Repeated single-pass radar DEMs provided by TanDEM-X are an excellent mean to detect drastic topographic changes, however such data is not systematically available at annual resolution.

In contrast to the detection of past events, detection of glaciers that may be prone to detach in the future is a much more

difficult task. On sk-17 and dp-19, increased crevassing could be only seen in high resolution images few of weeks prior to the detachment. This makes it extremely difficult to identify possible instabilities sufficiently early, especially when a glacier is not inspected on a regular basis. Similar, the Aru glaciers also showed increased crevassing just a few weeks before their detachments (Kääb et al., 2018). Indeed, even the supposedly tell-tale crevasses don't always reliably predict a detachment. For example, a small glacier near the Gulyia-Ice cap in the western Kunlun Shan has been showing detachment-like crevasses

since early 2018 (Leinss et al., 2019), but has remained stable so far, likely due to the stabilizing effect of its very broad tongue. Another option for an early identification of possible detachments would be through automated near real-time mapping of velocities using very high resolution sensors. However, based on our experience the detached glaciers in the Petra Pervogo range are too small for current optical or radar sensors like S1, S2 or Landsat to provide reliable velocity estimates. Increased data bandwidth and imaging capabilities of future sensors and high-repeat rate DEM differencing satellites could provide the

required data for early detection of possible detachments. In the specific catchments of this study, where we large mass flows



**Table 3.** Characteristics of the glaciers where we identified detachments. For glacier identification, only the most recent detachments are used as column title. For a full list see Table 1.

| Measure | sk03+06 | sk-17 | sk-16b | dp-19 | sha-19 |
|---|---|---|---|---|---|
| Glacier length (m) | 1080 | 1590 | 1100 | 1350 | 400-700 |
| Glacier width (m) | 250 | 270 | 350 | 300 | 130 |
| Aspect | N | NW | NE | NE | E |
| Lowest point (m) | 3410 | 3310 | 3550 | 2862 | 3450 |
| Highest point (m) | 3820 | 3900 | 4100 | 3400 | 3800 |
| Mean slope (°) | 22.3 | 21.8 | 30.0 | 23.5 | 24.9 |

occur frequently, in situ observation by radar or cameras could very likely act as warning systems to inform local population in time.

## 5.2 Detachment characteristics and triggers

Fundamentally, the question of which events to classify as glacier detachments - failures of low-angle valley glaciers that involve substantial amounts of the glacier - is a tricky one when the observations are purely based on remotely sensed imagery. In our study region, the task is further complicated by wide spread debris cover, which makes it hard to delineate glaciers. While the boundaries of the glacier detachment category are certainly fuzzy, we have classified nine of the fourteen detected events listed in Table 1 as certain or likely glacier detachments. We based out classification on satellite imagery and classified glaciers located in a valley or at least in a topographic depression as detachment when major parts of the ice volume detached. The posterior analysis of the detachment events shows that all share the characteristic low to medium surface slope of the detached area (15-25°) and that all occurred in a location where the GLIMS database (Raup et al., 2007) indicated the presence of a glacier. Remarkably, all events presented in this study happened within a roughly 30 km radius and the glaciers in the catchment areas present very similar characteristics regarding elevation and aspect (Table 3), with the SK catchment, for which the GLIMS data base lists five separate glaciers, appearing to provide particularly favorable conditions.

Henceforth, we focus our discussion on these events, in particular on the largest detachments sk-17 and dp-19. In comparing these two events with other detachments described in literature (in particular Aru, Kolka-Karmadon and Flat Creek), we find similarities in slope, lithology and the time of year of the events. Both images and the described lithology (sedimentary) suggest that the easily erodible bedrock and soft sediments are abundant in our study area. Similar to Kolka glacier, dp-19 was below a steep headwall and detached at the Bergschrund, so that the resulting mass movement involved basically the entire glacier.

As has been reported for other glacier detachments (Kääb et al., 2018; Gilbert et al., 2018; Jacquemart and Loso, 2019), there is a remarkable proximity, or in some cases overlap, between detaching and surging glaciers. Like others, we identified hundreds of surging glaciers throughout the Pamir, and the spatial distribution of the surging glaciers identified in our study is similar to Goerlich et al. (2020, Fig. 6). By comparison of the spatial distribution of surging glaciers with the rock types according to the geological map by Ibrohim et al. we found that surging glaciers occur predominantly in regions with soft and fine-grained rock-types. It is noteworthy, though the importance and effect not yet well understood, that the glaciers that





later detached (sk-17 and dp-19 in out study, but also the Aru and Kolka glaciers) exhibited a slightly steeper slope and were relatively short compared to their non-detaching surging neighbors (Fig. 10). Both sk-17 and dp-19 have surged in the past, but neither were in the midst of a surge immediately before their detachment, nor did they show any surge-like behavior. They did, however, show a significant acceleration in the weeks prior to the detachment. Therefore, we do not believe that sk-17 or

dp-19 were the consequence of a "runaway surge", but that both glacier surging and glacier detachments are favoured by a soft sedimentary bedrock. We rather conclude that the detachments were triggered by external drivers: because velocities increased during or after snowmelt, we suspect that increased liquid water input played a crucial role in lubricating the glacier base or saturating the underlying glacier bed (Gilbert et al., 2018). This idea is supported by the fact that all detachments happened in summer (July–September), when more liquid water is available and it's influence on the glacier dynamics is greater. We did

not find any obvious indication that earthquakes could have triggered the detachments. Instead, we have observed that 12 out of 14 mass movements, including the eight out of nine detachments, occurred in years when the mean annual air temperature was above the long-term trend.

The fact that relatively short and steep glaciers (compared to their surging neighbours) show detachments could be related to the reason that short glaciers are more likely to have a more homogeneous slope compared to long glacier. When enhanced

melt water lubricates the homogeneous base of a short glacier it is much more likely to detach compared to a long glacier where lubrication might be a more local effect and could possibly init a surge-cycle when a sufficiently high mass imbalance is present.

All of the investigate events were very mobile, though at first glace, their mobility, characterized by an angle of reach of around $\alpha = 10 - 15°$, was lower than that of the events at Aru and Kolka ($\alpha = 5 - 8°$) (Huggel et al., 2005; Kääb et al., 2018).

The lower mobility can be partly explained by the smaller volume involved ( Petra-Pervogo: $1\ 9 \times 10^6\ \mathrm{m}^3$, the others 70–130$\times 10^6\ \mathrm{m}^3$). However, if we compute the ratio $V/L$ between detachment volume and runout distance, the ratio is one to two orders of magnitude smaller compared to the Kolka and Aru detachments, indicating an extremely high mobility. This could be a consequence of the path geometry, which channelized the avalanches over a very long distance in a small area. The valleys of easily erodible sediments provided few obstacles and thus small energy loss. In addition, we think the exceptionally long

run out of 19.1 km of the debris flow event sk-16b, which angle of reach of $\alpha = 7.7°$ is comparable to the other large events, is caused by entrainment of the ice-water-sediment mixture deposited in the catchment by the sk-16a event five weeks before. A video of the event shows that the debris flow is almost as liquid as water (Radio Ozodi, 2016).

## 6 Conclusions

In this study we build an inventory of glacier detachments which occurred in the western Petra Pervogo range in Tajikistan.

Compared to a handful of other large glacier detachments around the entire world we found a cluster of at least nine, relatively small detachments within a radius of 30 km. The fact that multiple detachments occurred under very similar conditions (elevation, aspect, size, meteorological conditions) allows for studying external driving factors which can trigger the detachment of a valley glacier. We found that detachments occur in years with temperature above the long term trend, indicating that



with rising temperatures more detachments can be expected and that climate change has an direct impact on the occurrence of glacier detachments. Despite being a seismic active region, no immediate effect of earth quakes could be observed in our study site. Similar to other detachments, the glaciers in our study rest on a bedrock of soft sediments. We found that the entrainment of sediment-ice debris mixture from a previous ice avalanche of relatively small volume five weeks before caused an extraordinary long mud flow of 19.1 km. We also observed a spatial correlation between the occurrence of surging glaciers in the Pamir mountains and soft, fine-grained sediments. However, we did not observe that the studied glacier detachments were a consequence of surging but we think that soft sediments are a prerequisite for detachments and at least a favouring factor for hydrologically controlled glacier surging. This is supported by our observation that detachments occurred predominantly in summer after snow melt and in years with above-trend temperatures. From the fact that detached glaciers are shorter and steeper compared to surging glaciers in the same region we hypothesize that melt water penetrating to the glacier base can lubricate major parts of the relatively small bedrock of soft sediments which then can lead to detachment of the entire glacier, especially if the glacier is relatively steep and the destabilized area is not supported by a stabilizing tongue of smaller slope. In contrast, for longer glaciers it is unlikely that the entire glacier loses friction at the bedrock and it might instead be more likely that the glacier shows a temporary surge-like advance.

*Code and data availability.* Copernicus Sentinel-2 data and USGS Landsat 8 data were processed by ESA and were downloaded the Sentinel hub with the EO Browser: https://www.sentinel-hub.com/explore/eobrowser/. L1–7, ASTER, and Sentinel-1 data were processed using the Google Earth Engine (Gorelick et al., 2017) using Java scripts available on request from the authors. Declassified Keyhole imagery is available from the NASA USGS Earth explorer https://earthexplorer.usgs.gov/. TanDEM-X data is available on request from DLR https://tandemx-science.dlr.de/ and was provided by the proposal leinss_XTI_GLAC6600. DigitalGlobe data were provided by the Commercial Archive Data for NASA investigators (cad4nasa.gsfc.nasa.gov) under the National Geospatial-Intelligence Agency's NextView license agreement. The facilities of IRIS Data Services, and specifically the IRIS Data Management Center, were used for access products used in this study. IRIS Data Services are funded through the Seismological Facilities for the Advancement of Geoscience (SAGE) Award of the National Science Foundation under Cooperative Support Agreement EAR-1851048.





# Appendix A: Additional imagery of detachments and avalanches

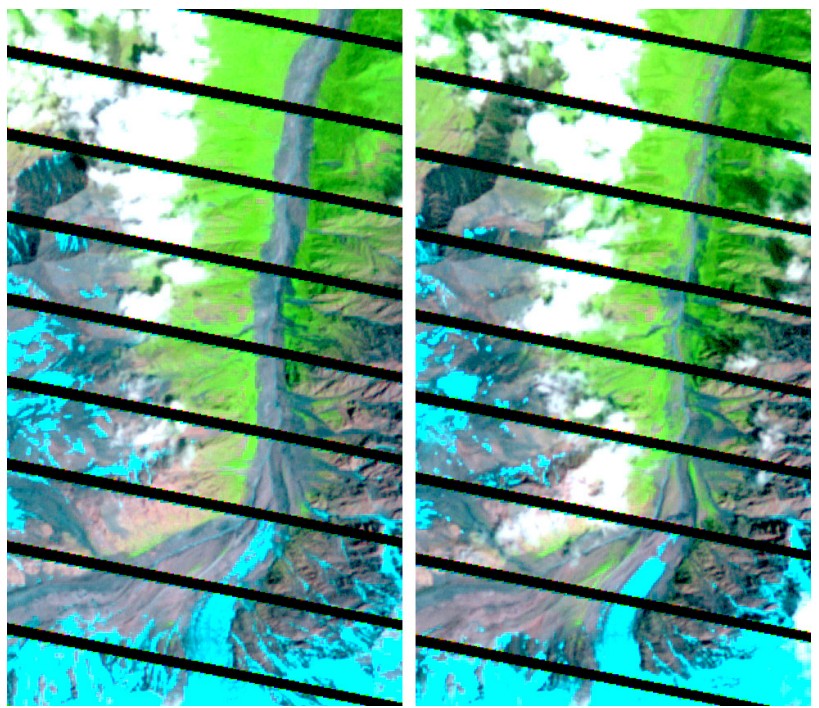

**Figure A1.** L7 false color images (Band 7,8,3 = SWIR, pan, red) from 07 July 2004 and 01 August 2013 show that vegetation on the Kolka-Karmadon rock-ice avalanche has recovered within about 10 years. The stripes are due to the failure of the scan line correlator of L7 in 2003. Landsat-7 image courtesy of the U.S. Geological Survey.

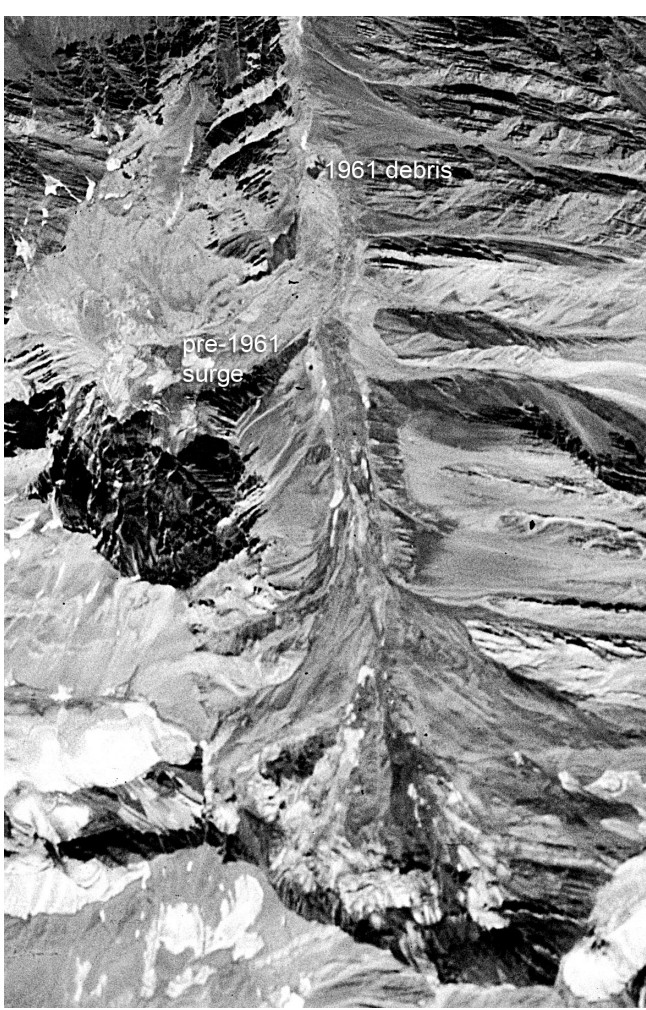
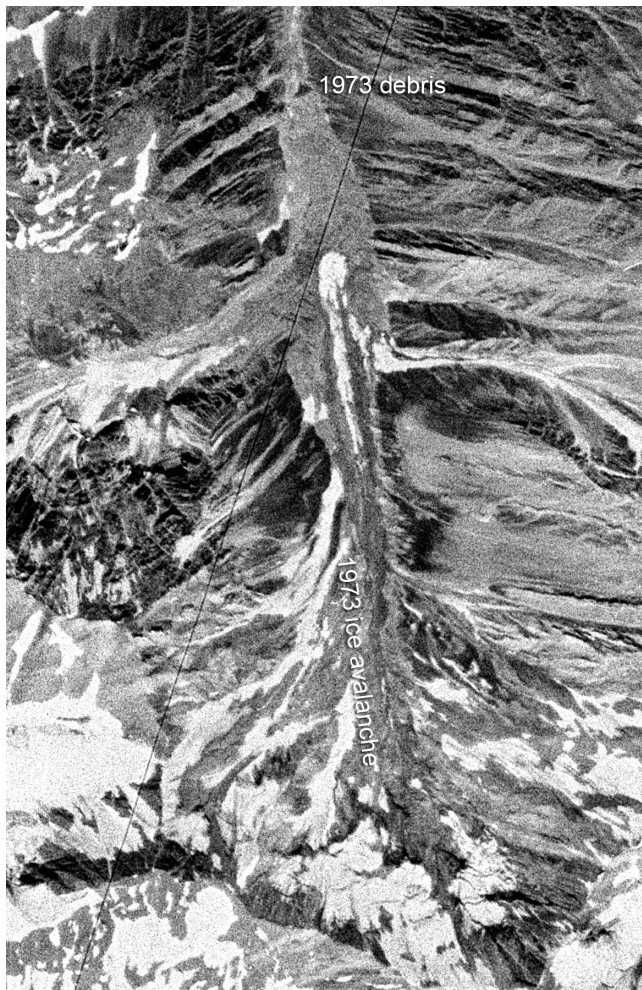

**Figure A2.** Keyhole images from 30 August 1961 and 03 August 1973 of the SK catchment area. In (a) the tongue of a pre-1961 surge from the western tributary is indicates the debris front. In (b) the debris front has advanced, the valley bottom appears widened, is filled with more debris and a new ice-avalanche covers the valley bottom and parts of the pre-1961 surge front. The visible avalanche looks very similar to the sk-06 avalanche in Fig. A3b. (a) Keyhole-3 image (DS009023023DV206_206_d) and (b) Keyhole-7 image (DZB1206-500080L018001), courtesy of the U.S. Geological survey.


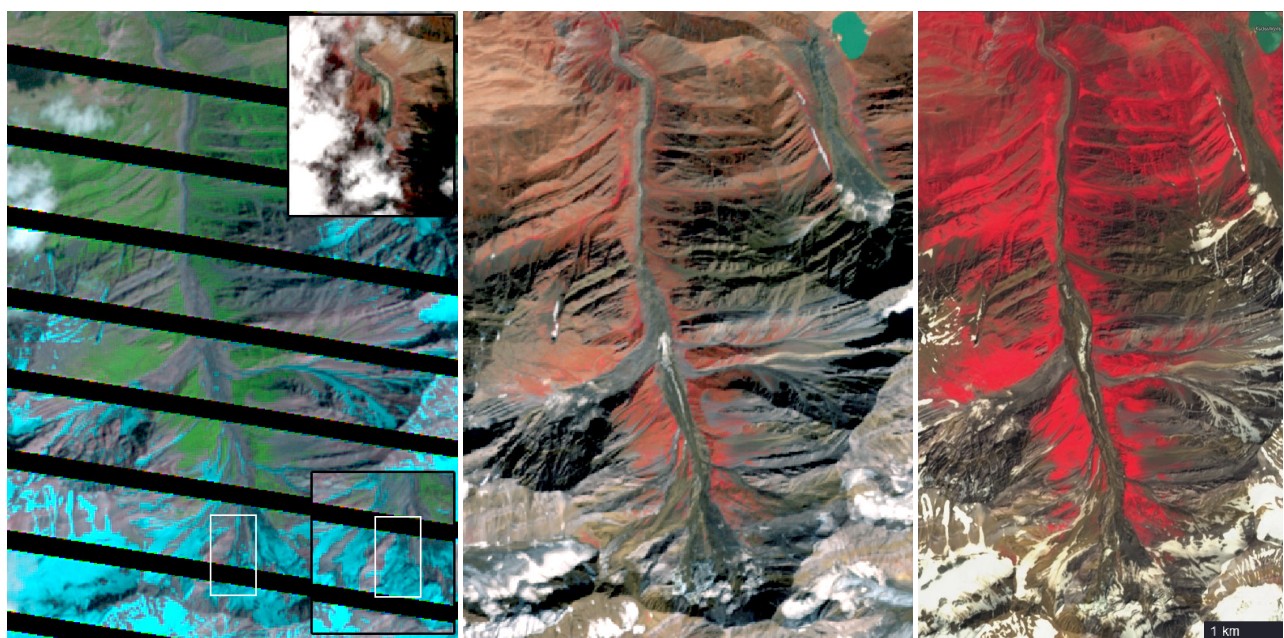

**Figure A3.** (a) Image from 10 August 2004 showing the missing glacier in the white box (lower inset from 24 August 2003). The end of the ice avalanche sk-03 is shown in the upper inset (25 September 2003), (b,c) the avalanches sk-06 (image from 08 September 2006) and avalanche sk-16a (image from 26 July 2016) originated from the same location as sk-03. Landsat-7 and ASTER image courtesy of the U.S. Geological Survey. Copernicus Sentinel Data (2020).

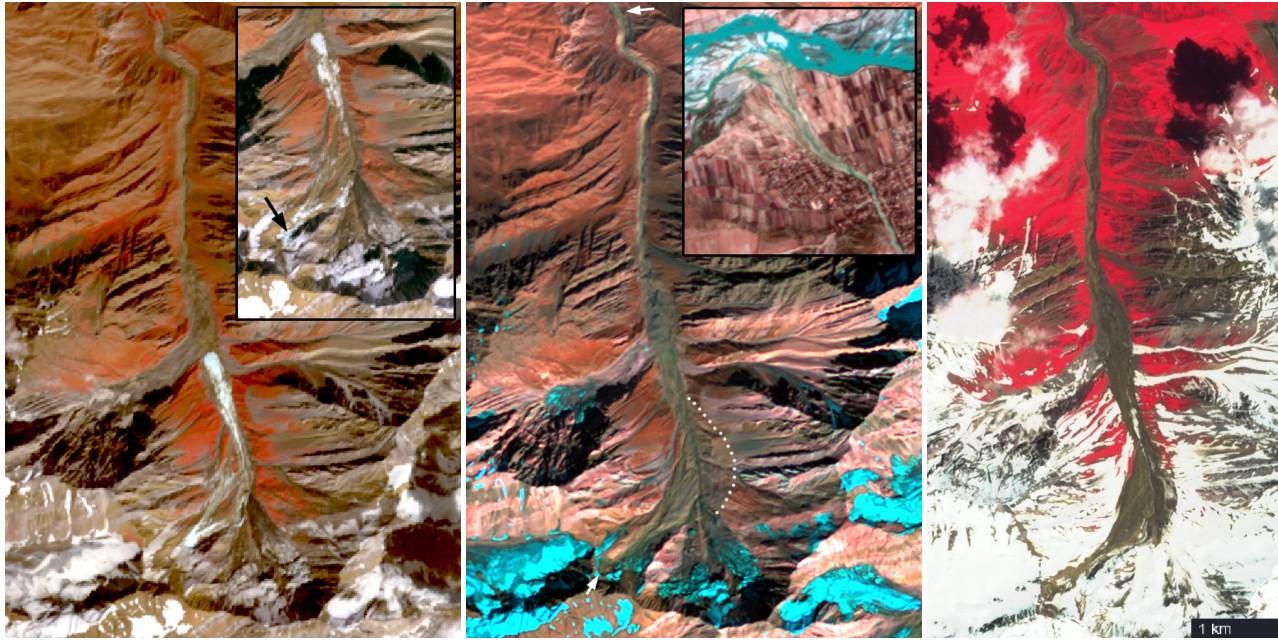

**Figure A4.** (a) image from 18 September 2004 (inset: 04 October 2004) of the sk-04 detachment, (b) The trim line of the sk-16b detachment is indicated by dots in the L8 image from 20 September 2016. One arrow indicates the rupture line of the detachment and the other arrow (at the top of the image) trees removed by the resulting debris flow. The inset shows the alluvial fan 19 km downstream where the mud flow reached the Surkhob river. (c) images from 01 July 2019 of the sk-19 ice/rock avalanche. ASTER and Landsat-8 imagery courtesy of the U.S. Geological Survey; Copernicus Sentinel Data (2020).




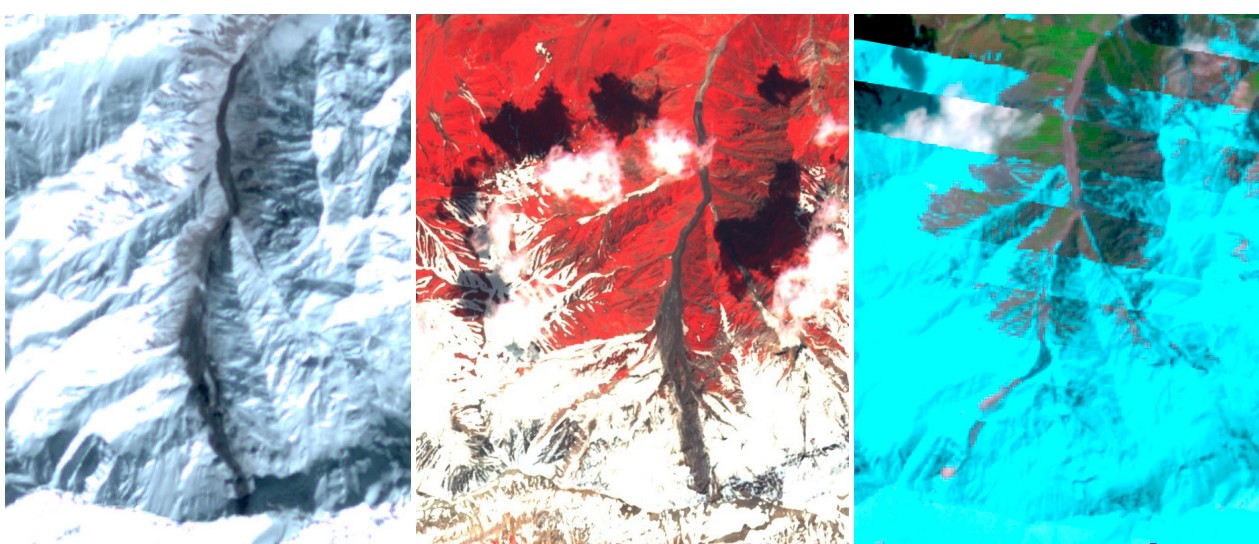

**Figure A5.** (a) image from 18 March 2001 of the shi-03 event, (b) image from 21 June 2017 of the (second) shi-17 rock/ice avalanche, (c) image from 11 and 27 May 2009 of the shi-09 avalanche. Landsat-7 image courtesy of the U.S. Geological Survey; Copernicus Sentinel data (2020).



*Author contributions.*  SL, EB, MJ jointly wrote the manuscript, SL processed the TanDEM-X data and wrote the Google Earth scripts, SL, EB analyzed the data, MJ computed the World View DEMs differences and calculated uncertainty estimates, MD provided relevant local
440   information, initiated the seismic study, and indicated two of the detachments, SL coordinated the study.

*Competing interests.*  MJ was funded through a NASA Earth and Space Science fellowship. The authors declare that they have no conflict of interest.

*Acknowledgements.*  The authors thank Mike Willis and Brie Corsa for troubleshooting and running the SETSM processing and Irena Hajnsek for her support and valuable discussion of this study.



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
