# Peer review of "Glacier detachments and rock-ice avalanches in the Petra Pervogo range, Tajikistan (1973–2019)"

_Natural Hazards and Earth System Sciences, 2020_

## Referee Comment (RC1) · Anonymous Referee #1 · 29 Oct 2020

General Comments:

The authors present new data and observations on the locations and characteristics of glacier detachments in Tajikistan. Given the lack of worldwide data on these types of events this is a valuable dataset. The authors used an impressively large amount of imagery and DEM data to conduct their search. However, this paper lacks key information on the methodology used to identify and classify detachment and other mass flow events. The authors provide valuable comparisons of event conditions and geometry (glacier slope, bedrock geology, surging, etc.) and provide some comparisons to other detachments worldwide. However, the conclusions drawn about climate change and temperature are largely unsubstantiated given that the inventory of detachment events was collected from data with varying resolution and quality, and may therefore

be biased towards more detections with higher quality (recent) imagery.

Specific Comments:

1. The terminology used for glacier detachments and other mass flow events should be addressed near the beginning of the manuscript and clarified throughout. Several terms including "glacier detachments," "mass flows," and "avalanches" are used interchangeably throughout the paper and at points it is difficult to understand whether you are referring broadly to all types of studied events or to specific types of events. In Table 1 you further classify events as detachments, ice avalanches, or rock/ice avalanches, yet this classification scheme is not consistent with the ways that these events are addressed throughout the paper. I would recommend adding a section to the introduction that describes the dichotomy of event types which are addressed throughout the paper. For example, overall you are studying mass flow events, which can be further classified as either glacier detachments, ice avalanches, or rock/ice avalanches, etc.

2. An improved description of the methods used to identify and classify detachments and other events is needed. Please provide more details on how you identified events. Was this done with a manual search, an automated search, etc.? Did you attempt to distinguish between detachment events and rock/ice avalanches or were both event types combined in your search. If they were separated, what was your criteria with which you were able to differentiate events? At what scale did you conduct your search? What is the minimum event size that you were able to detect using your technique?

3. It seems likely that your inventory is biased by the quality of available imagery since you used 15 m resolution imagery for recent years, but much larger resolution for older events. This is a reasonable approach for the purposes of detecting as many events as possible given the available data, but you cannot justify the conclusions you draw about temperature trends with an inventory that is likely biased by data availability/quality.

4. I would recommend improving the structure of the paper in several areas. The

methods section is very disjointed by all of the subsections discussing measurements of event characteristics. You could potentially combine many of these subsections into a more coherent description of data attribute collection, which would also leave room for an expanded discussion of your detection and identification techniques. I would also recommend removing or shortening the portions of the Results section that highlight the smaller events that are not discussed further. Highlighting them with reference to Table 1 may be sufficient. Instead I would focus more on the major events which you have more data for and include in the discussion.

5. Consider focusing more on the strengths of your observational data and emphasizing your analysis of detachments in relation to surging glaciers, bedrock geology, glacier slope etc. I think these aspects of your results and discussion are very interesting and important to emphasize given the overall lack of data on these types of events throughout the world. It would also be interesting to expand your discussion/comparison with other documented events.

6. Technical language, in particular the use of vague or ambiguous terms, should be improved. I have attempted to make suggestions for several of these instances in the following section. Grammatical errors and typos, as detailed in the following sections, should also be addressed.

Technical Corrections:

L2: Suggest removing "the" so the phrase reads "Common to all known cases are large..." L7: What type of temperature data are you referring to with the phrase "the past 46-years trend"? Are you referring to mean air temperatures, maximum temperatures, minimum temps? Average annual temperatures, etc.? The statement is vague as is. L7: Suggest rewording to "No active glacier surges were observed immediately before detachments..." L8: Insert "digital" preceding "elevation model" L9: Unclear what is meant by "pronounced" in this context. Would "preceded" be a better description here? L18: Unclear why there is a section labeled 1.1 here since there are no other

sub-sections in the introduction. The use of "Glacier detachments" as a heading does not fit with your discussion of multiple different types of events (surges, rock avalanches etc.) in this section. L21: Remove "a" preceding "relatively low" and would recommend changing "low slopes" to "low-angle slopes" L22: Recommend changing to "velocity increases by one or two orders of magnitude, but detachment does not occur." L23: What is meant by "favored by a climatic envelope"? L24: Should rock avalanches also be included if you are discussing events that initiate from headwalls? L25: Change to "For both types of mass flows.." L26: Remove "in additionally" L27: Suggest changing to "increase in liquid water content, making the resulting mass flows, which sometimes transform into debris or…" L29: Replace "potentially reaching inhabited areas" with "which increases the potential for such events to reach inhabited areas." L35-36: Please check grammar. L37: Replace "for all of the probably best-studied" with "for many well-studied events, including…" L46: Remove "of a series of" L48: Recommend changing to "and the surging history of individual glaciers." L49: You switch from present ("analyze") to past ("investigated") tense. Choose one and be consistent. L58-59: Based on journal standards, is this the correct citation format for in-line citations? L60-61: Elsewhere you use "runout" as one word. L61: Insert "the" preceding "two largest detachments" L72: Word missing near the end of the line? L79: Year for Ibrohim et al. reference? L89: Recommend changing to "with an increase of almost 1 degree C in fall and winter" L90-91: Check use of "run out" versus "run-out". Elsewhere you use "runout." L102: Suggest changing to "at resolutions of" L108: Suggest changing to "No imagery from" L111: "green bands" L116: Should rock avalanches also be included in this? At some points throughout the text you distinguish between rock and ice avalanches, while at others you seem to use the terms "ice avalanche" and "rock avalanche" interchangeably. Please clarify your terminology and make it clear to readers whether your use of "ice avalanche" encompasses both rock and ice (and rock-ice) avalanches or is distinct. L118: check use of "run out" as two words. L118-119: How did you distinguish the runout zone from the source (detachment) zone? L128: Please write out the full name prior to introducing an acronym. L129: "to estimate" L132: How did the DEM differencing reveal a previously unknown event? What criteria was used to classify this as an event? L134: "estimated following" L134-135: Please check reference formatting for in-line references. L135-136: Are clouds the only cause for large differences in your images? Please discuss other potential sources of error. What about snow cover? What is the intrinsic error of individual DEM layers based on processing etc.? L142: Was this only done for the events that you know are detachments or also for rock/ice avalanche events? If so, how were detachments distinguished from the other event types? I would work to clarify your terminology surrounding detached glaciers versus rock/ice avalanches and make it clear whether you are using the overall term for both types of events. L147: Please check grammar. L152: Earthquakes is one word. Please check here and elsewhere. L152: What was your screening criteria for determining whether an event was triggered by an earthquake or not? L170: Are you using "mass flows" as an encompassing term for both detachments and rock/ice avalanches? If so, please define your use of terminology earlier in the text so it is clear to readers what event types you are talking about here. L191: Remove one instance of "until" L197: Remove "leading" L203: See notes on clarification of terminology. L211: What was your criteria for "unusual crevassing"? L212: How do you quantify "heavy crevassing"? L232: Spelling of "loosing" L287: Specify what you mean by "long-term trend" L290: Here and elsewhere please correct the use of "earth quakes" as two words. L314: Please check grammar. L329-330: What do you consider "large" in the context of your study area. Please specify. L335: In reference to your point about vegetation being missing in strongly eroded valleys, what about areas/ecosystems/elevations that would not be expected to sustain vegetation in the first place? L336-337: Please explain what you mean by "separation of snow and clouds." Based on the Landsat channels, how do you distinguish between snow and clouds? Do they appear texturally different? Different colors? L342: Correct "single-pass" and change "mean" to "means" L345: Correct "images a few weeks prior" L355: A word seems to be missing from the end of the line. L362-363: Here you briefly describe the criteria used to distinguish glacier detachments from other types of mass flow events.

[Figure]

I would recommend moving this discussion to the methods section and expanding on it in more detail. This is a critical piece of how you collected your results, but has yet to be mentioned in the text. L363: Correct "out" L381: Typo "out" Line 398: Change to "investigated events" L415: "earthquakes" is one word

Figure 1: Include North arrow on map. Can you specify what you mean by "modified" in the Figure 1 caption?

Table 1: - What do you mean by "other events"? Please clarify in the Table caption. - This table distinguishes between detachments, ice avalanches, rock avalanches, and questionable detachments, however you do not indicate in the methods section how you distinguished between each event type. Additional discussion of this process is needed in the methods section. - Furthermore, I would recommend clarifying the terminology you use interchangeably for the different event types that you discuss throughout the paper. It seems like in the broadest sense, you are discussing mass flows, which can be classified as either detachments, ice avalanches, or rock/ice avalanches. To make this clearer, I would clarify your terminology throughout the text and for example, use "Mass Flow Type" instead of "Avalanche Type" in the Table header. - How did you distinguish between release and runout areas? This should be explained more thoroughly in the methods section. What does "yes" in parentheses mean under the "surge observed" category? - What does your Slope measurement correspond to? - Check use of "run out" as two words.

Figure 3: The presence of increased crevassing in a) and b) is not visible at this scale. Can you show a more detailed image or indicate what area of the image you observe the crevassing in? Is increased crevassing not expected normally throughout the season? How do you distinguish between snow melt which exposes more crevasses and an otherwise significant increase in crevassing?

Figure 4: Is this the same area that is shown in Figure 3 (with the exception of Figure 3d)? If so, it would be helpful to indicate such with a common point or lat/long ticks.

Can you point out what you mean by "strong crevassing" in part d?

Figure 5: Suggest changing first line of caption to "locations of"

Figure 6: This figure shows the glacier detachment sufficiently, but there is also overall snow/ice melt between the pre- and post-event images. Could you compare the final "detachment" image with an image from the previous year around the same time period (July) to show what the area looked like prior to the detachment, but with similar snow conditions?

Figure 9: - In the caption "cyan" for temperature at the Lyairun station should be corrected. - Replace "earth quakes" with earthquakes. - Please clarify the use of black "bullets" versus black "dots." Instead you could use "large" and "small" black dots.

Figure 10: Figure appears blurry, resolution should be improved.

---

## Author Comment (AC1) · 9 Nov 2020

Dear Reviewer 1,

We highly appreciate your constructive review and thank for the quick response allowing for an interactive discussion. In this interactive response, we like to clarify the key points adressed by you. We will provide a more extensive answer in final comments at the end of the discussion phase.

**General Comment 1**: "this paper lacks key information on the methodology used to identify and classify detachments and other mass flows".

**Answer**: In section 3.1, we wrote: "To identify and characterize detachments and

ice avalanches we analyzed [list of sensors] (. . .). To characterize (. . .) we compared consecutive images or images from different years acquired during the same time of the year".

Though we roughly explained what we did, we agree that the information *how* detachments and ice avalanches were identified and *how* they were classified is missing. Below we provide this information:

We did all detection and classification based on manual inspection of coregistered image series. The scale of identifiable events was given by the sensor resolution, but we considered only events, which showed a total length of at least 2 km length.

We identified events by looking for obvious traces of large mass flows, like clearly visible avalanche patterns in the valleys, removal of vegetation, and changes in surface color indicating overtopping of landscape by mass flows.

After identification of such an event, we inspected the release zone for local losses of ice or rock volumes. Despite using mainly optical imagery, we identified volume losses in a qualitative way by the clearly visible holes in the mountain slopes, often casting shadows, which were not visible a few days before (or on a similar day in the year before). In some cases, we identified glacier detachments by a sudden, significant loss of ice cover.

For classification of the events, we followed the description of Evans and Delaney (2015) as outlined in the introduction section. We classified events as glacier detachment when we could clearly identify a hole in a previously glaciated valley showing the newly exposed glacier bedrock which remaining after the loss of ice masses. In addition, we checked whether the avalanche debris showed considerable amounts of ice. We classified events as ice avalanches when the detached ice was not located in a clearly visible valley or when no exposed bedrock was visible. To distinguish the events from other types of avalanches, we checked whether the avalanche debris contained large amounts of ice (the infrared channel allows for separation of snow and ice). We classified events as ice-rock avalanches when the release area was at least partially ice covered, the avalanche deposit showed traces of ice and rock, and when the release area was not located at a valley bottom. We classified events as rock-ice avalanches when satellite imagery indicated that rock fall run over an at least partially glaciated area.

**General Comment 2**: "(...) the conclusion about climate change and temperature are largely unsubstantiated given that the inventory of detachment events was collected from data with varying resolution and quality, and may be biased towards more detections with higher quality (recent) images".

**Answer**: We fully agree that the inventory is very likely biased towards more recent events where a larger number of different satellites are available and where a much higher spatial, radiometric and temporal resolution is available. However, we like to stress that we think to *compensate for this bias*, when comparing in Fig. 9 the temperature of the year of the event with the *linear trend* of the temperature in the past 46 years and find that most events happen in years with *above-trend* temperatures.

Because of the observation bias, we *do not intend* to draw any conclusion from the fact that the more frequent observation of more recent events shows a (very likely pseudo-)correlation with warming temperatures due to climate change.

Specific comments:

**Comment 1)**: Usage of terminology.

**Answer**: We fully agree. Though we start the introduction (1st paragraph) with a definition of glacier detachments and also of ice avalanches, we will try to make the difference more clear ( -> add dichotomy of event types to introduction). Will refer to

this section when describing how we classification the different events (See general comment 1).

**Comment 2)** Improve description of method, scale, classification.

**Answer**: We agree with all suggestions. We will consider them as suggested in the answer to general comment 1.

**Comment 3)** Bias of inventory.

**Answer**: We fully agree with the fact that the inventory is biased toward more events in recent years. However, as detailed in general comment 2, we think that our conclusion is not biased because of the comparison to the temperature trend.

**Comment 4)** Improving structure of method and result section.

**Answer**: Thanks you very much for your constructive suggestions. We will try to structure the method section according to data attribute collection. In the results, we will try to shorten the sections about the smaller events where possible and will refer instead to table 1. We might provide more details about streamlining the structure as soon as we have received the second review.

**Comment 5)** Emphasize analysis of detachments in relation to surging glaciers, bedrock geology, glacier slope.(. . .) Expand your discussion to other documented events.

**Answer**: We would like to point to the following, recently published discussion paper (Kääb 2020) which provides a large review, comparison and analysis of most detachment events identified up to date: https://tc.copernicus.org/preprints/tc-2020-243/. As we consider our paper as an inventory and documentation of the events, which hap-

pened in the Petra Pervogo Range in Tajikistan, we think that a larger comparison would be beyond the scope of our paper.

**Technical correction + comment 6:**

**Answer**: We absolutely appreciate the detailed feedback and especially the constructive suggestions. We will consider all technical corrections in the revision.

With best regards, The authors,

S. Leinss, E. Bernardini, M. Jacquemart, and M. Dokukin

---

## Referee Comment (RC2) · Anonymous Referee #2 · 15 Dec 2020

General Comments

This study shows new insights into the detachment of glaciers and its processes in general. They discovered several detachments in the north-western Pamir and described them in detail. The authors provided many new information about the environments where this high number of detachments took place. Further, they used many different datasets to complete their research but missed more detailed description of their methods to detect the glacier dynamics because clear definitions are missing. Of course, it is difficult (similar to e.g. glacier surges) to explicitly define a glacier detachment and to distinguish from other mass movements.

Even though the variability of datasets is high, there might be more scenes of declassified data (KH-4a/b or KH-9) which have a high resolution up to ∼2m and could give

further insights into glacier states in the past (1960 - 1980). This data would also show glacier surface structures like crevasses. This could also improve the lack of high-quality data from early Landsat missions. Another idea to improve the variability of datasets is to analyse Russian topographic maps. The maps at least in 1:100.000 are available online and maybe it is possible to find 1:50.000 as well.

I like the idea of DEM differencing to detect glacier changes but how reliable are the DEM datasets? Especially in steeper regions (and that is one of your cases) the DEMs show quality issues. What about other DEMs? You would get more difference images to better investigate changes over shorter periods.

Your results section is very difficult to follow. Please make the structure clearer. Maybe it would make sense to combine the results and discussions section so that the reader can better connect the different cases.

Detailed comments

L18 subheading not necessary as there is no second heading within the introduction L20 delete Âńa Âż and Âńaround Âż L35 two time "which" Fig1 credits of the background image are covered by the map insert. L61 "…with the run out of the two largest…" L78 sandstone L85-89 it is not clear which numbers belong to which station…make it clearer, e.g. "respectively" L102 same as for L85 L110 KH-3 data? You do not mean KH-4/Corona data? There might be other datasets of Corona, especially KH-4B with resolutions up to 1.8m and Hexagon? L118 calculate L134 following L142 TDX might be a more common abbreviation for TanDEM-X L147 stations L149 temperature L161 delete "is" Fig3 maybe add glacier outlines, maybe mark the heavily crevassed areas L178 maybe show the two images from 2nd and 3rd August 2019? L188 Are you sure that it reached this height? Which type of baserock/sediment is situated there? Is it possible that just the slope slided down when the lowest parts were scratched off by the ice flow? L191 delete "until" L196 indicate L199 "…3d and are visible…" L246 make sure whether you use the term "run out" or "runout" in general L249 reported

L270 how can the glacier be not existent but build up mass until 2019? Fig9 caption "...black bullets indicate earth quakes..." L299 "...there was a large number..." L314 delete "it" L355 delete "we" L379 add year for "Ibrohim et al." L414 "...change has a direct impact..."

---

## Author Comment (AC2) · 26 Jan 2021

Final response letter

Adressing comments by anonymous Referee #2, including references to comments and answer by Referee #1.

We thank both reviewers for their timely response, for carefully scrutinizing the manuscript, and for the time spent on the manuscript and for their valuable comments to improve the paper. Here, we first address the general comments of both referees, followed by a list of brief technical corrections and brief responses to detailed comments.

General comment 1, Referee #1: "(..) this paper lacks key information on the method-

ology used to identify and classify detachment and other mass flow events. (. . .) I would recommend adding a section to the introduction that describes the dichotomy of event types which are addressed throughout the paper." in agreement with the general comment 1 of Referee #2: "The authors (. . .) missed more detailed description of their methods to detect the glacier dynamics because clear definitions are missing"

Author response 1: We totally agree with both reviewers and will start the paper with an introductory description/dichotomy of different types of catastrophic mass flows (CMFs) following (Evans and Delaney, 2015). Further, we will add a paragraph to the method section 3.1 where we describe how we distinguished the observed CMFs using optical imagery (For details see answer to general comment 1 in AC1 the interactive discussion). Following that, We will try to structure the subsequent method sub section according to data attribute collection (as suggested by the specific comment 4 of Referee #1). We will also thoroughly check the entire paper for consistend usage of the terms detachment, ice/rock/rock-ice avalanche and glacial debris flows. The above points are in agreement with answers to specific comments 1), 2) and 4) of referee #1 in the interactive discussion.

General comment 2, Referee #1: "the conclusions drawn about climate change and temperature are largely unsubstantiated given that the inventory of detachment events was collected from data with varying resolution and quality, and may therefore be biased towards more detections with higher quality (recent) imagery."

Author response 2: As already outlined in the interactive response to Referee #1 (general comment 2), we fully agree with that our inventory is very likely biased towards more recent events where a larger number of different satellites are available, and where a much higher spatial, radiometric and temporal resolution is available. To support this, we will add a graphic (attached) which visualized the increasing number of satellite imagery. However, we like to stress that we think to be able to compensate for this bias, because in Fig. 9 we compare the annual mean air temperatures of the events with the temperature from the linear trend of annual mean temperatures of

the past 46 years. We do not draw our conclusion from the recent increase in event observations but from the fact that most events happen in years with above-trend temperatures. We will clarify this in the conclusion and will mention that despite of an observational bias of the inventory, the comparison of the annual mean temperature with the long term annual temperature trend allows for the conclusion that during rising temperatures, where frequently new record temperatures are observed, more detachments can be expected. (Answer in agreement to specific comment 3 of referee #1 in interactive discussion).

General comment 3, Referee #2: "there might be more scenes of declassified data (KH-4a/b or KH-9) which have a high resolution up to 2m and could give further insights into glacier states in the past (1960 - 1980). This data would also show glacier surface structures like crevasses. This could also improve the lack of highquality data from early Landsat missions."

Author response 3: We have checked the USGS Earth Explorer dataset and found in total 22 acquisitions from KH-3, KH4, KH4a/b, and KH-9 dating back untip 1961-08-30. In only seven of them, the surface is well visible. Other imagery shows dense clouds or too much snow cover. In the high resolution imagery, we found in the entire area strongly crevassed glaciers but, except for the 1973 event, no clear evidence of major mass flows. In some valley, missing vegetation could origin from past mass flows but these could also be simply erosion patterns. We will have a closer look at this imagery but do not expect much additional information.

General comment 4, Referee #2: "Another idea to improve the variability of datasets is to analyse Russian topographic maps. The maps at least in 1:100.000 are available online and maybe it is possible to find 1:50.000 as well."

Author response 4: There seem to be some inofficial sources of Russian topographic maps found on vlasenko.net. However, on https://maps.vlasenko.net/soviet-military-topographic-map/ only the scale 100'000 seems to be available. Examples

for the PetraPervogo Range are: https://maps.vlasenko.net/smtm100/j-42-034.jpg and https://maps.vlasenko.net/smtm100/j-42-046.jpg. Scale 50'000 is not provided for the region of interest. However, as these maps have are derrived products, they don't provide any precise enough information about the status of glaciers or topography.

General comment 5, Referee #2: I like the idea of DEM differencing to detect glacier changes but how reliable are the DEM datasets? Especially in steeper regions (and that is one of your cases) the DEMs show quality issues.

Author response 5: See line 134 where we wrote: "Volume uncertainties associated with all DEM differences were estimated follow the method described in (Miles et al., 2018).". Based on our experience with radar DEMs, slopes oriented in the azimuth direction (north/south) can be observed quite well. As all glaciers for which we studied DEM differences show an exposition towards north, they are not affected by layover or shadow. Using optical imagery (WorldView) we studied the DEM differences only for slopes lower than 30 degree where no significant quality issues are present. We will add the information to the method section that "derived volumes were estimated only on glaciers with slopes lower than 30 degree" (See also Table 1 and 3).

General comment 6, Referee #2: What about other DEMs? You would get more difference images to better investigate changes over shorter periods.

Author response 6: We are not aware that other DEMs with specific time stamps and sufficient quality are available for the region.

General comment 7, Referee #2: "Your results section is very difficult to follow. Please make the structure clearer. Maybe it would make sense to combine the results and discussions section so that the reader can better connect the different cases." We will follow the suggestion in the specific comment 4 of Referee #1: "I would also recommend removing or shortening the portions of the Results section that highlight the smaller events that are not discussed further. Highlighting them with reference to Table 1 may be sufficient. Instead I would focus more on the major events which you have

more data for and include in the discussion."

—

All technical corrections / detailed comments will be considered as listed below:

Answers to technical corrections from referee #1: - L2: typo corrected. - L7: annual mean temperatures. - L7: rephrased this and the following sentence to "Digital elevation model (DEM) differences indicate a surge-like behavior about 10 years before the two largest detachments, but different to other detached glaciers, one glacier retreated before detachment while the other remained stagnant before increased sliding pronounced the impending detachment." - L8: Inserted "Digital" in the above sentence. - Unclear what is meant by "pronounced" in this context. Would "preceded" be a better description here? – we meant "signalized" but as increased velocities do not signalize a detachment, preceded is the better choice. - L18: subsection label 1.1 removed. - L21: Removed "a"; changed "low slopes" to "low-angle slopes". - L22: reformulated as suggested. - L23: We mean: "Glacier surges are favoured by an envelope of climatic conditions (sevestre 2015). - L24: Yes, we will distinguis and include rock avalanches (observed in the Shikorchi catchment). - L25: replaced by "For all three classes, potential energy is transformed..." - L26: replaced by ", and also additionally entrained sediments, " - L27: suggestion accepted. - L29: accepted "which increases the potential for such events to reach inhabited areas." - L35-36: removed "which". - L46: Removed "of a series of". - L48: Changed as suggested. - L49: Chose present tense. - L58-59: Corrected citation format for in-line citations according to journal standard "by Kääb (2020)". L60-61: we checked that "runout" is used consistently throughout the manuscript. - L61: Inserted "the". - L72: Missing word "events" added. - L79: Year for Ibrohim et al. reference? Unfortunately, we neither found a year nor received a response to a possible copyright request to reproduce the map. - L89: wording changed as suggested. - L90-91: run-out replaced by runout. - L102: change irrelevant as we will replace the paragraph by a graphic and table. - L108: Wording changed as suggested. - L111: typo corrected. - L116: Should rock

avalanches also be included in this? At some points throughout the text you distinguish between rock and ice avalanches, while at others you seem to use the terms "ice avalanche" and "rock avalanche" interchangeably. Please clarify your terminology and make it clear to readers whether your use of "ice avalanche" encompasses both rock and ice (and rock-ice) avalanches or is distinct. – Will be done as sugessted in the general author response 1. - L118 done. - L118-119: How did you distinguish the runout zone from the source (detachment) zone? – We did not distinguish and mapped the entire avalanche area. We will clarify this. - L128: Please write out the full name prior to introducing an acronym. – As we do not use this acronym further in the text, we replaced "using SETSM (ref)" by "using the SETSM-algorithm (ref)" - L129: typo corrected. - L132: How did the DEM differencing reveal a previously unknown event? What criteria was used to classify this as an event? – The DEM difference revealed a negative height change of a suspicious shape. Using optical satellite imagery, we could classify the event according to the described classifiers. - L134: typo corrected. - L134-135: inline reference formatting checked. I think "in (author+year)" is correct, and also "by author (year)". But not "in author (year)" and also not "by (author year)". - L135-136: Are clouds the only cause for large differences in your images? Please discuss other potential sources of error. What about snow cover? What is the intrinsic error of individual DEM layers based on processing etc.? – Yes, clouds were the only cause for large DEM errors. We will still mention whether snow cover posed a problem and will mention intrinsic DEM errors. L142: Was this [studying DEM differences] only done for the events that you know are detachments or also for rock/ice avalanche events? If so, how were detachments distinguished from the other event types? I would work to clarify your terminology surrounding detached glaciers versus rock/ice avalanches and make it clear whether you are using the overall term for both types of events. – This was only done for the two largest detachments. For differentiation between different events, see suggestion in answer to general comment 1. L147: grammar corrected. L152: "Earthquake" used throughout the paper. L152: What was your screening criteria for determining whether an event was triggered by an earthquake or not? – As written, we checked for earthquakes (> mag 5, within 14 days of event and within 100 km radius). Then we discuss smaller earthquakes (> mag 4.5) within a closer temporal/spatial proximity to the events. L170: Terminology of mass flow, detachment, rock/ice avalanche will be defined earlier in the text. L191: typo corrected. L197: changed "leading and" to "leading to". L203: terminology will be clarified. L211: replaced "unusual crevassing" by "Crevasses, surrounding the detaching area, become increasingly visible". L212: removed "heavy crevassing" L232: typo corrected. L287: long term trend -> 46 year trend. L290: "Earthquake" used throughout the paper. L314: Grammar corrected. L329: large = Mass flows with runouts larger than two kilometers or wider than 50 - 100 m. L335: In reference to your point about vegetation being missing in strongly eroded valleys, what about areas/ecosystems/elevations that would not be expected to sustain vegetation in the first place? – It depends on how fast any surface reaches its "equilibrium color" after perturbation by an external event. L336-337: how do you distinguish between snow and clouds? - Clouds are at a higher elevation and show therfore stronger scattering of SWIR radiation which is absorbed by moisture in the atmosphere before reaching snow cover. L342: typos corrected. L345: typo corrected. L355: sentence corrected. L362-363: Here you briefly describe the criteria used to distinguish glacier detachments from other types of mass flow events. I would recommend moving this discussion to the methods section and expanding on it in more detail. This is a critical piece of how you collected your results, but has yet to be mentioned in the text. – Will be moved to methods in accordance to the general comment 1. L363, 381; typos corrected. L414: "Earthquake" corrected. Figure 1: We will add a north arrow and specify what we modified (borders and labels added). Table 1: other events = glacier avalanches, rock avalanches and rock-ice avalanches. Events will be distinguished according to the general comment 1. Table 1: What does "yes" in parentheses mean under the "surge observed" category? – It means, in the past, but not directly before the detachment, a surge-like dynamics was observed. Table 1: What does your Slope measurement correspond to? – It's the slope of the detached area.

Figure 3: The presence of increased crevassing in a) and b) is not visible at this scale. Can you show a more detailed image or indicate what area of the image you observed the crevassing in? - Arrows added. Figure 3: Is increased crevassing not expected normally throughout the season? How do you distinguish between snow melt which exposes more crevasses and an otherwise significant increase in crevassing? – The location of the crevasses does not match with crevasses exposed by melt. We will discuss this in the caption/text.

Figure 4: Is this the same area that is shown in Figure 3 (with the exception of Figure 3d)? If so, it would be helpful to indicate such with a common point or lat/long ticks. – It's not exactly the same area. We will either add some common points or lat/long ticks. Figure 4: Can you point out what you mean by "strong crevassing" in part d? - strong crevassing at the glacier outline.

Figure 5: suggestion accepted.

Figure 6: Could you compare the final "detachment" image with an image from the previous year around the same time period (July) to show what the area looked like prior to the detachment, but with similar snow conditions? – Yes, we will add such an image.

Figure 9: color corrected. Used "large and small black dots" as suggested. Figure 10: Resolution will be improved.

Answers to detailed comments from Referee #2: - L18, L35: typos corrected. - Fig1 credits of the background image are covered by the map insert. -> Credits are also in the caption. - L61: missing word added. - L78, typo done. - L85-89: added "respectively" (two times) - L102: the paragraph will be replaced by a figure and table showing acquisition times, number of acquisitions per year and bands/resolution used for each satellite. - L110 KH-3 data? You do not mean KH-4/Corona data? – No, KH-3 is correct. See USGS documentation https://www.usgs.gov/centers/eros/science/usgs-eros-archivedeclassified-data-declassified-satellite-imagery-1?qt-science_center_objects=0#qt-science_center_objects. There might be other datasets of Corona, especially KH-4B with resolutions up to 1.8m and Hexagon? – Yes, see authors response 3. - L118,134: typos corrected. - TDX might be a more common abbreviation for TanDEM-X. - Might be true. As TanDEM-X is a satellite formation consisting of two satellites, TerraSAR-X and TanDEM-X, I prefer to use TSX and TDX to refer to the individual satellites and use TDM to refer to the formation of both satellites. This is also consistent with the L1B Product naming convention, therefore we prefer to keep the abbreviation TDM. - L147, L149, L161: typos corrected. - Fig3: maybe add glacier outlines; maybe mark the heavily crevassed areas. – We will add arrows to where crevasses are opening and will change the caption to "From (a) to (b) crevasses opening around the glacier outline are visible (arrows)." - L178 maybe show the two images from 2nd and 3rd August 2019? - These two images show a partial cloud cover. Instead, we show images from 28 July and 07 August which reveal the pre- and post-detachment situation more in detail. - L188 Are you sure that it reached this height? Which type of baserock/sediment is situated there? Is it possible that just the slope slided down when the lowest parts were scratched off by the ice flow? – As the apparent flow pattern indicate a very continous line along the topography, and because the flow line crosses over several gully in the topography, we conclude with a high certainly that this height was reached. The slope shows very little pattern where material could have slid down and we could not identify any new erosion gullies or eroded arcs indicating slides triggered by scratched off material. In this curve, shown by the arrows in Fig. 3d, the left slope of the valley is not very steep (max. 35 degree) making it again unlikely that the slope slided down. Comparison of high resolution Google Earth Imagery from 2016-05-31 vs. 2019-08-20 does not indicate any slope with slide down. - L191, L196: typos corrected. L199: Sentence rephrased, comma added. L246: we made sure to use "runout" in general. L249: typo corrected. L270 how can the glacier be not existent but build up mass until 2019? – In the valley where this glacier was located, we did not identify any ice in 2013. However, it seems that at this location ice or snow

(and very likely large amounts of sediments) are building up and form a multi-year ice body showing crevasses. As the glacier has the GLIMS ID G070995E39014N we consider it as a glacier. Fig9 caption, L299, L314, L355: typos corrected. L379 add year for "Ibrohim et al." : Unfortunately, we could not find any year for this reference. L414: typo corrected.
* * *
**Fig. 1.**

---

## Author Response (AR1)

Dear editor, Dear reviewers,

Here we submit the major revision of the manuscript nhess-2020-285 where we considered all suggestions by the reviewers and as already detailed in the final response to the discussion.

We have made major changes to the manuscript to address the reviewers major concerns. These are:

- We start the introduction section with a clear classification and description of the events.
- We added a rigorous description to our method section and moved data about satellites acquisitions to the corresponding figures and tables to improve the readability of the method section.
- We have restructured the result section according the the suggestions of reviewer 1, starting with a description of the major events, followed by a list of the minor events.
- We have removed the general conclusions about climate change and have replaced it by a more objective description our our results where we observe that most of the events occur in years with above-trend temperatures. We explicitly mention that our dataset is biased towards more recent years to prevent any biased conclusion from the fact that we observe more frequently events in the most recent years.

Due to the large number of changes made to the manuscript (see changelog file), and because the editor suggested that it will be reviewed again, we summarized here only the major changes made to the manuscript and like to refer for detailed point-by-point-responses to our final response in the discussion forum.

With kind regards,

Silvan Leinss